# DelvePO: Direction-Guided Self-Evolving Framework for Flexible Prompt Optimization

## Abstract

Prompt Optimization has emerged as a crucial approach due to its capabilities in steering Large Language Models to solve various tasks. However, current works mainly rely on the random rewriting ability of LLMs, and the optimization process generally focus on specific influencing factors, which makes it easy to fall into local optimum. Besides, the performance of the optimized prompt is often unstable, which limits its transferability in different tasks. To address the above challenges, we propose **DelvePO** (**D**irection-Guid**e**d Se**l**f-E**v**olving Framework for Fl**e**xible **P**rompt **O**ptimization), a task-agnostic framework to optimize prompts in self-evolve manner. In our framework, we decouple prompts into different components that can be used to explore the impact that different factors may have on various tasks. On this basis, we introduce working memory, through which LLMs can alleviate the deficiencies caused by their own uncertainties and further obtain key insights to guide the generation of new prompts. Extensive experiments conducted on different tasks covering various domains for both open- and closed-source LLMs, including DeepSeek-R1-Distill-Llama-8B, Qwen2.5-7B-Instruct and GPT-4o-mini. Experimental results show that DelvePO consistently outperforms previous SOTA methods under identical experimental settings, demonstrating its effectiveness and transferability across different tasks.

## 1 Introduction

The rapid advancement of Large Language Models (LLMs) (DeepSeek-AI, 2025; Li et al., 2025) has revolutionized various real-world applications (Shao et al., 2024; Zheng et al., 2025) . Prompt, a method that steers LLMs to produce desired results without modifying parameters, has garnered significant interest among non-AI experts from different domains (Wan et al., 2024; Guo et al., 2025; Fernando et al., 2024). Consequently, the rapid growth in users has increased demand for prompt engineering methods.

Previous efforts primarily focused on manually designing specialized prompts (Brown et al., 2020; Kojima et al., 2022; Wei et al., 2023). However, this kind of method is time-consuming and demands extensive trial and error, making it less versatile for diverse tasks and limiting their real-world effectiveness. To reduce the human effort required for constructing effective prompts, many researches (Shum et al., 2023; Wang et al., 2023c; Zhang et al., 2022; Feng et al., 2024; He et al., 2024) have increasingly explored methods such as curating unified demonstrations for related tasks, systematically designing domain-specific templates, and identifying critical factors for prompt performance. However, these methods exhibit limited applicability across diverse scenarios.

Subsequently, a series of research emerged that employ optimization algorithms to refine prompts. Such approaches (e.g. APE (Zhou et al., 2023b), PromptBreeder (Fernando et al., 2024), and Evo-Prompt (Guo et al., 2025)) synergistically integrate the efficiency inherent in the algorithms with the powerful text processing ability of LLMs, achieving relatively stable performance enhancement on target datasets. Although these studies analogize the mutation operation in evolutionary algorithms to the rewriting operation of LLMs, they fail to fully harness the efficiency and rapid convergence inherent to such algorithms, which ultimately limits the realization of their performance advantages in prompt optimization. The primary reason lies in the inherently stochastic nature of the evolutionary process: the directionality of mutation operations remains uncontrollable, while their interpretability is also notably limited. Furthermore, these methods neglect the potential impact of

constituent components within a prompt on overall performance, leading to premature convergence in local optima. For example, during evolutionary phase of EvoPrompt, the initial prompt inherently contains the "role" as a critical component. However, due to the stochastic nature of the mutation process, the stochastic mutation process may accidentally remove this component. Once discarded, it cannot be reintegrated into subsequent evolutionary iterations. Such degradation significantly heightens the risk of premature convergence in local optima. A parallel limitation is observed in the PromptBreeder method, which exhibits even higher stochasticity, as its implementation not only uses two distinct mutation prompts but also employs diverse mutation operators, amplifying randomness throughout the optimization process. We summarize that a robust Prompt Optimization (PO) must have the following characteristics:

- **Seamlessly integrating domain expert experience**: For tasks in different domains, prior experience from domain experts can be incorporated into the PO algorithm, thus improving the efficiency of the optimization process.
- **Actively exploring factors that may affect prompt performance**: The method can actively explore factors affecting prompt performance to guide optimization using historical data.
- **Adaptively identifying optimal prompts for different LLMs with varying performance**: The algorithm self-adjusts to discover the best prompts for target tasks across differently specialized models and scenarios, ensuring broad applicability in diverse professional contexts.

Integrating insights from existing research, we propose **DelvePO** [1] (**D**irection-Guid**e**d Se**l**f-E**v**olving Framework for Fl**e**xible **P**rompt **O**ptimization) that adaptively accommodates diverse LLMs and self-improves through guidance from its historical optimization strategies. Inspired by the concept of Loci (the corresponding location of genes with important functions) and Alleles (different versions of the same gene) on genetics, this framework first decouples prompt instructions into functional components (analogous to Loci). Subsequently, it iteratively evolves these components by exploring the potential impacts of diverse allele variations, ultimately achieving holistic optimization of the entire prompt through systematic recombination. In particular, building upon the components, we introduce working memory mechanism (i.e., Component Memory and Prompt Memory) to guide the evolutionary process. Component Memory is designed to capture evolutionary trends in individual components and utilize these trends to guide further optimization of each element. Take the component a step further, Prompt Memory creates interconnections between components by utilizing contextual information to guide the progressive optimization of the entire prompt. The contributions of our work can be summarized as follows:

- To the best of our knowledge, our work is the first to introduce memory mechanism to guide prompt optimization, not only stabilizing the performance of the entire prompt population but also greatly reducing the time required for evolutionary operations.
- By decoupling prompt into multiple components and designing guided evolutionary mechanisms, our framework integrates multiple influencing factors into a single prompt. This integration not only enhances the scalability of PO methods but also improves the interpretability of the optimization process, significantly lowering the difficulty to interact with the system.
- For LLMs with varying performance levels, our framework can elicit their capabilities, striking a good balance between exploring diverse components and exploiting the current derived good components, ultimately obtaining optimal prompts that adapt to the target tasks and LLMs simultaneously. Extensive experimental results on multiple datasets and three widely-adopted LLMs reveal that DelvePO outperforms manually crafted prompts and existing PO methods.

## 2 PRELIMINARIES

Given task $T = (\boldsymbol{D}, \boldsymbol{A})$, $\boldsymbol{D}$ is the task-related dataset and $\boldsymbol{A}$ represents the corresponding answer to the dataset, prompt optimization can be briefly described as follows: Guided by the working memory mechanism, the initial prompt population $\boldsymbol{P}_{init} = \{p_1, p_2, \cdots\}$ is continuously optimized to obtain the final prompt population $\boldsymbol{P}_{final}$. The best prompt $p^*$ can be selected as follows:

$$p^* \leftarrow \underset{p \in \boldsymbol{P}_{final}}{\arg\max} f_{eval}\big(\phi^{\mathcal{LLM}}(p, \boldsymbol{D}_{dev}), \boldsymbol{A}\big)$$

---

[1]DelvePO is available at https://anonymous.4open.science/r/DelvePO

where $\boldsymbol{D}_{dev}$ is the development dataset and $\phi^{\mathcal{LLM}}(p, \boldsymbol{D}_{dev})$ means that the prompts and questions are combined and then fed into the LLM to produce the corresponding response. The important concepts used in our proposed framework are described below.

**Components** Similar to the relationship between Loci and Chromosome, components are mainly used to identify the location of key factors that affect task performance in prompts. Different tasks can introduce distinct components or reuse existing ones. Components are extensible, i.e., the type and number of components can be user-defined, and our method can also evolve synchronously as the context length that LLMs can receive increases. In this paper, we construct a comprehensive and representative component pool from a broad set of related literature. Further details on how the components are studied and predefined in our framework are provided in Appendix E.

**Templates** To bind components to prompts, we design a general template (corresponding to the Chromosome functionally), whose content is mainly composed of two parts: general and unchanging text; domain-specific and replaceable descriptive text (i.e., components and their corresponding values). For the descriptive text, its main functions include explaining domain-specific components, connecting different components, and providing contextual semantics about components. To overcome the instability of LLMs in recognizing components, we borrow the design idea of "markup" from HyperText Markup Language (HTML) to define different domain components. Taking "**\<role>\</role>**" as an example, the "role" is one of the various component types. Accordingly, the value of the component will be enclosed within the markup pairs, i.e., \<role>**Sentence Simplifier**\</role>. More details can be found in Figure 6 in Appendix F.

## 3 METHODOLOGY

### 3.1 FRAMEWORK OF DELVEPO

Our self-evolution prompt optimization framework consists of 4 necessary functional modules: Sampling & Update module, Task-Evolution module, Solution-Evolution module and Memory-evolution module. We define the **Task** as "discover the promising direction of evolution", that is, determining the component (types or values) that need to evolve in the next step under the guidance of components memory. We define the **Solution** as "make sure the process of evolutionary operation and perform evolutionary operation", i.e., under the guidance of prompts memory, evolutionary operations are applied to the component values according to the selected evolution type: for a single sample, only mutation is performed, while for two samples, both mutation and crossover are executed. For memory-evolution, it mainly uses the evolved prompts and component value pairs before and after evolution to update the prompts memory and components memory, respectively. In the sampling and update module, when the number of iterations reaches a pre-defined value, the population is updated. Otherwise, a new sampling operation is performed within the current population, which in turn triggers the next round of self-evolution operations. The designs of **DelvePO** framework is shown in Figure 1. Next, we first introduce the working memory mechanism.

**Components Memory** stores the corresponding component values before and after evolution, which is selected according to the mutated component type. The value pairs will be ordered by descend, i.e., when injecting to the final prompt, the first value performs better than the second. Components Memory will guide the selection of components in the Task-Evolution stage.

**Prompts Memory** stores the prompts after each step of evolution. The evolved prompts are stored in descending order according to their performance scores. There are two forms of prompts memory: discrete form and continuous form. The discrete version only stores discrete combinations of component value in the prompt. And the continuous version stores a complete prompt formed by injecting component value into the template, which means that it stores continuous text containing context. Prompts memory will be used to guide the mutation of component or the crossover of the prompt in the Solution-Evolution stage.

### 3.2 OVERVIEW OF DELVEPO

As shown in Figure 1, the workflow of **DelvePO** contains several core stages as outlined below.

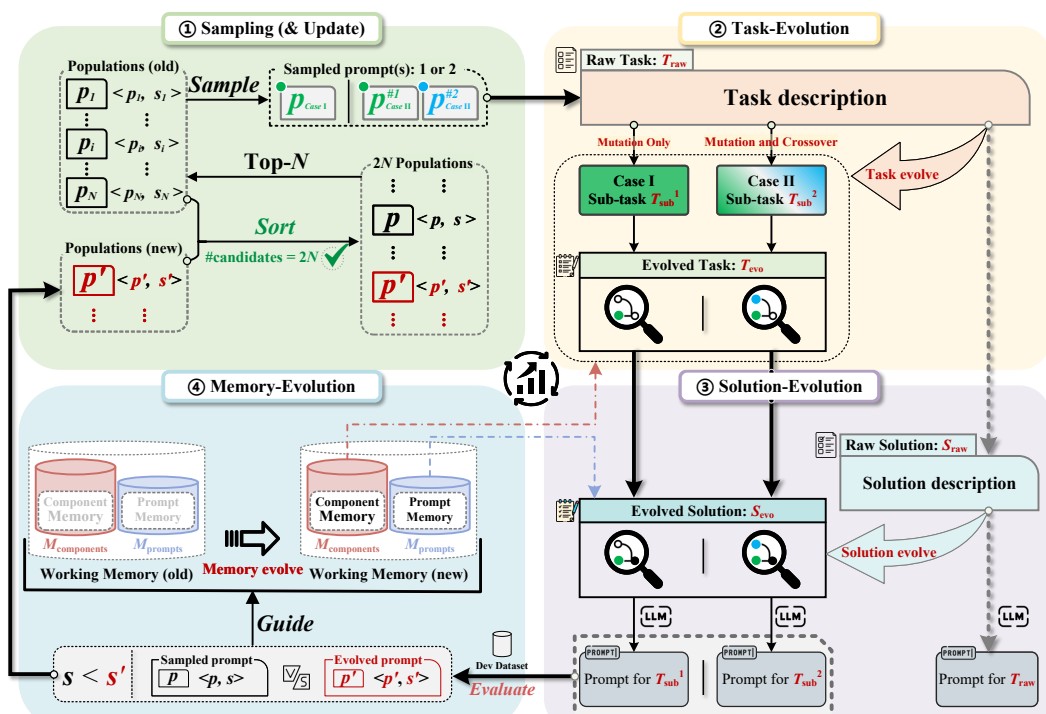

Figure 1: **The Framework of DelvePO**. Initialization begins with predefined components, which are concatenated to form individual $p$; multiple individuals constitute the initial population **Populations (old)**. At each step, one individual (Mutation only) or two individuals (Mutation and Crossover) are sampled, and the **Sub-task** determines the evolutionary direction (i.e., the mutated component type). Guided by **Task-**, **Solution-**, and **Memory-Evolution** modules, selected prompts are iteratively evolved, contrasting with unguided optimization. The new population **Populations (new)** is accumulated across epochs, and once the threshold is reached, the population is updated to initiate the next round of self-evolution.

**Initialization & Sampling**: First, we use task-agnostic component-value generation prompt (see Figure 4 in Appendix C) to generate candidate values for each component type. Then, we randomly sample from these candidates and inject the selected values into the population-construction template (illustrated in Figure 6 in Appendix F) to construct the initial population. Each individual in the initial population is evaluated on the development dataset to obtain its performance score. Finally, the sorted population is stored as the initial prompts memory. Before the population evolves, there is no components memory. After initialization, the sampling process begins, aiming to select prompts from the current population for evolution. Inspired by genetic principles, there are two main ways to generate new individuals: mutating a single individual or performing crossover between two individuals. Notably, mutation may also occur during crossover. To account for these cases, we assume that the number of individuals selected in each sampling step can be either 1 or 2.

The evolutionary process mainly includes two parts: generating new individuals based on selected individuals; generating and storing the working memory. Specifically, there are 3 types of evolution, namely **Task-Evolution**, **Solution-Evolution**, **Memory-Evolution**. The mechanism of Task-Evolution and Solution-Evolution is shown in Figure 2.

**Task-Evolution** For task evolution, considering the components and the evolutionary operations (mutation and crossover), we design two kinds of evolutionary sub-tasks. The detailed information is shown in Figure 8 and Figure 9 (see Appendix G).

- **Sub-task I**: This task mainly uses mutation operations to process a single candidate prompt. First, the semantic comprehension capability of the LLMs is utilized to obtain the relevant insights of component evolution from the component memory $M_{\text{components}}$. Then, the insights are used to guide the selection of components. Finally, the selected components will be treated as the promising direction to guide the evolution of mutation-based solution.

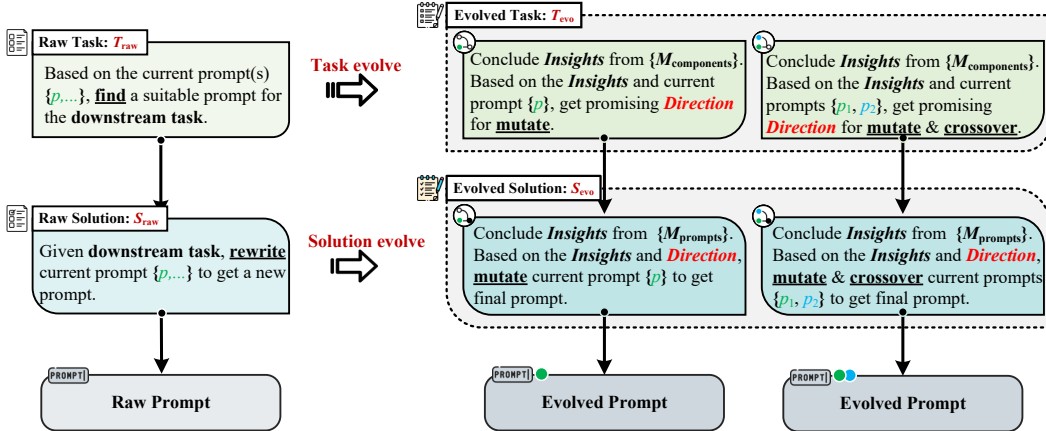

Figure 2: The mechanism of Task-Evolution and Solution-Evolution. *Using the pseudo-prompt to explain the details of Task- and Solution-Evolution.*

- **Sub-task II**: After performing Sub-task I on the two candidate prompts, we can get the respective component types set $C_1$ and $C_2$ for two prompts (say $p_1$ and $p_2$) as the promising direction for mutation. The final mutated component type is selected as $\hat{C} = C_1 \cap C_2$. Next, for each component in $\tilde{C} = C \setminus \hat{C}$ where $C$ denotes the set of all component types, corresponding contents from $p_1$ and $p_2$ are extracted to construct a pair. Then, based on the insights derived from $\boldsymbol{M}_{\text{components}}$, one value from each pair is selected as the potential value to improve performance of the prompts after evolution. Finally, the component types from $\hat{C}$ will be treated as the promising direction to guide the evolution of crossover-based solution, and the selected values from $p_1$ or $p_2$ whose component types coming from $\tilde{C}$ will also be passed into the corresponding Solution-Evolution phase to help construct the final prompts.

**Solution-Evolution** The main goal of solution evolution is to utilize the insights (derived from the prompts memory) and direction (received from the task-evolution) to perform evolution operations on the corresponding content in the current prompt and generate a new prompt that performs better. In this phase, we propose 2 sub-solutions corresponding to 2 sub-tasks. Depending on whether the prompt is continuous or discrete, each sub-solution can also be further divided to eliminate the effect of prompt format on the final result.

- **Sub-solution I**: Extract component contents from current prompt based on the results obtained by sub-task I (i.e., the mutated components that are most likely to improve prompt performance). The extracted contents are then mutated using insights obtained from the prompts memory $\boldsymbol{M}_{\text{prompts}}$ stored in discrete or continuous forms. Those contents that have not been mutated will be retained in new prompts. Finally, the mutated and unmutated component contents will be integrated as the result of sub-solution I. The corresponding prompts are shown in Figure 10, 11 (see Appendix H) for the prompts memory in discrete and continuous forms, respectively.
- **Sub-solution II**: This mainly uses the results from sub-task II as a guide, and extracts component contents from the currently selected two prompts. And the evolutionary operations would combine mutation and crossover. First, for components that do not require mutation, the corresponding content is received from sub-task II. Then, for the component that need to be mutated, we extract its content from the two prompts. Based on the evolutionary insights derived from the prompt memory $\boldsymbol{M}_{\text{prompts}}$, the mutation operations are performed on the extracted content. Next, the generated two prompts will crossover on the component types that need to be mutated. Finally, the results obtained from the mutation and crossover operations are integrated to generate a new prompt as the result of the sub-solution II. The details are shown in Figure 12 for the prompts memory in discrete form and Figure 13, 14 for continuous form (see Appendix H).

**Memory-Evolution** is based on the component pairs and prompts both before and after the evolution to update the corresponding components memory and prompts memory, which is used to guide the next evolution process. In this module, the **evaluation** will be performed. Specifically, to clearly describe the evaluation process, we illustrate a general form of a prompt designed for LLMs that can be applied across different tasks (shown in Figure 7). Evaluation refers to calculating the per-

formance score of the generated new prompts on the development dataset based on the evaluation metrics of the target task, according to which components can be sorted and memory can be updated.

**Update**: Add the evolved prompts to the temporary population generated in each iteration. When the iteration ends, the temporary and current populations are mixed, and Top-N is selected as the updated population for the next iteration based on performance.

The details of **DelvePO** are outlined in Algorithm 1, which can be found in Appendix C.

## 4 EXPERIMENTS

### 4.1 EXPERIMENTAL SETTINGS

**Baselines** In our experiments, We choose 6 commonly used methods which have been widely proven to be efficient in the field of prompt optimization as our baselines, which are: Crafted by human experts, CoT-ZS, CoT-FS, Promptbreeder, APE, and EvoPrompt.

- **Human** corresponds to manually crafted prompts by experts, as detailed in the relevant literature Zhang et al. (2024); Sanh et al. (2022), which primarily derived from prior studies.
- **CoT** has been extensively applied in various domains, represents a rationale-based approach. We evaluate two representative forms of CoT: **CoT-ZS** (Zero-Shot CoT, Kojima et al. (2022)) and **CoT-FS** (Few-Shot CoT, also known as Manual-CoT, Wei et al. (2023)).
- **APE** (Zhou et al., 2023b) regards instructions as programs and uses Monte Carlo Search to select appropriate instructions as optimized prompts under LLM guidance.
- **Promptbreeder** (Fernando et al., 2024) further investigates the effect of different mutation strategies on self-optimization based on elaborately designed evolutionary operations.
- **EvoPrompt** (Guo et al., 2025) introduces evolutionary algorithms to prompt optimization for the first time. Considering different scenarios, it instantiates its framework using two practical evolutionary algorithms. According to its statement, compared with GA method, the DE method has a wider range of use in solving complex problems. Therefore, we select EvoPrompt-DE as our baseline, and denote it simply as EvoPrompt.

**Datasets and LLMs** To demonstrate the generalizability of our method, we conducted experiments on 11 datasets across three LLMs, covering diverse domains and representative real-world tasks. The details information about datasets and LLMs are represented in Appendix B. Other experimental details (e.g., Computational Resources and Hyperparameter Details) are represented in Section 6.

### 4.2 MAIN RESULTS

Following the same settings as baselines, we tested the best prompts obtained during training. The main experimental results (as shown in Table 1) on DeepSeek-R1-Distill-Llama-8B are reported as averages over three random seeds, with standard deviations provided. It is worth noting that we observed Promptbreeder to be significantly more time-consuming than other methods (as shown in Figure 3). To balance the diversity of baselines and ensure the fairness in training time, we therefore report results for Promptbreeder using a single random seed.

From Table 1, we can observe that our method achieves substantial improvements over manual approaches. Among the automated optimization methods, our method consistently outperforms baselines, demonstrating not only its effectiveness but also its adaptability to different task types. From the results on classical NLP benchmarks, we observe that the baselines perform well, confirming their effectiveness on established datasets. However, on more recently introduced benchmarks that demand broader capabilities, automated prompt optimization methods generally perform better, with our approach showing particularly substantial improvements. These results indicate that as LLMs continue to advance, prompt optimization techniques must likewise evolve, and our framework delivers consistently strong performance across diverse domains.

To further evaluate the performance of our framework on different LLMs, we conducted additional experiments across different task types on the closed-source model (GPT-4o-mini, results reported in Table 2) and the widely used open-source model (Qwen2.5-7B-Instruct, shown in Table 5 in Appendix D). The experimental settings were kept identical to the main experiments. As shown in the results evaluated on these two LLMs, our framework consistently delivers either superior

Table 1: Main results on different downstream tasks for DeepSeek-R1-Distill-Llama-8B. Since expert-written prompts are not available for all datasets, sign ("-") is used to indicate missing cases.

| Method | Classical NLP | | | Question-Answering | | Domain-specific | NLG | Avg. |
|---|---|---|---|---|---|---|---|---|
| | Subj | MR | CoLA | SQuAD | TREC | FinPB | SAMSum | |
| Human | 26.00 | 55.89 | - | - | 54.67 | - | 25.68 | - |
| CoT-ZS | 70.00 | 68.00 | 65.45 | 43.91 | 68.00 | 60.00 | 3.23 | 56.74 |
| CoT-FS | $\underline{83.00}$ | $\underline{90.67}$ | $\underline{70.63}$ | 47.92 | $\underline{71.00}$ | 68.67 | 4.25 | 62.81 |
| Promptbreeder | 35.00 | 86.00 | 55.58 | 54.16 | 60.00 | 59.00 | 27.88 | 51.20 |
| APE | $74.67_{(2.85)}$ | $83.67_{(1.67)}$ | $68.75_{(1.20)}$ | $67.57_{(1.62)}$ | $42.33_{(2.40)}$ | $70.67_{(2.33)}$ | $\underline{30.02}_{(0.85)}$ | 61.25 |
| EvoPrompt | $82.00_{(2.08)}$ | $83.00_{(1.00)}$ | $66.75_{(2.73)}$ | $\underline{68.17}_{(1.14)}$ | $67.00_{(1.53)}$ | $\underline{72.00}_{(1.53)}$ | $29.18_{(0.47)}$ | $\underline{65.55}$ |
| **DelvePO** | $\mathbf{83.67}_{(1.20)}$ | $\mathbf{91.00}_{(1.00)}$ | $\mathbf{76.25}_{(1.49)}$ | $\mathbf{68.53}_{(2.61)}$ | $\mathbf{76.00}_{(2.08)}$ | $\mathbf{73.33}_{(3.06)}$ | $\mathbf{32.05}_{(0.25)}$ | **70.48** |

Table 2: The results on different downstream tasks for GPT-4o-mini.

| Method | Classical NLP | | Domain-specific | Multi-domain | Avg. |
|---|---|---|---|---|---|
| | Subj | CoLA | FinPB | AG's News | |
| Human | 27.33 | - | - | $\underline{87.56}$ | 57.45 |
| CoT-ZS | 67.67 | 81.40 | 73.67 | 80.33 | 75.77 |
| CoT-FS | 82.00 | **84.93** | 80.67 | 83.00 | 82.65 |
| Promptbreeder | 45.00 | 67.72 | 72.00 | 78.00 | 65.68 |
| APE | $\underline{79.61}_{(1.78)}$ | $81.53_{(1.93)}$ | $94.93_{(0.78)}$ | $84.60_{(0.93)}$ | 85.17 |
| EvoPrompt | $76.70_{(1.90)}$ | $82.72_{(2.11)}$ | $\underline{96.97}_{(0.52)}$ | $86.50_{(1.40)}$ | $\underline{85.72}$ |
| **DelvePO** | $\mathbf{91.07}_{(1.03)}$ | $\underline{83.14}_{(1.90)}$ | $\mathbf{98.63}_{(0.62)}$ | $\mathbf{89.40}_{(0.81)}$ | **90.56** |

or competitive performance across multiple task types, demonstrating its robustness and general effectiveness when applied to diverse LLMs.

## 4.3 COST ANALYSIS

In our experiments, the overhead primarily stems from the training time required for open-source LLMs and the number of tokens consumed in API requests for closed-source LLMs. Accordingly, for DeepSeek-R1-Distill-Llama-8B, we randomly selected one dataset from each task type and measured the time cost of different baselines, with results presented in Figure 3. The statistics indicate that our method consistently outperforms or matches the baselines in terms of optimization speed, particularly when compared with PromptBreeder. This also explains why we report its results using a single random seed. Overall, the results demonstrate that our method can more effectively exploit the rapid convergence property of evolutionary algorithms for faster optimization.

Moreover, we reported token usage in terms of the actual monetary expenditure, as shown in Table 6. Overall, as shown in Table 2 and Table 6, although our method requires higher expenditure, it consistently delivers performance above or competitive with the baselines, indicating that our approach offers a favorable balance between performance and cost. We also analyzed the reasons behind the generally higher token usage. The primary factor is that the content stored in the memory module is included as part of the input provided to the target LLMs. In future work, we plan to integrate prompt compression techniques into the framework to reduce this overhead.

## 4.4 ABLATION STUDY

To evaluate the impact of the memory mechanism in our framework, we conducted ablation experiments on GPT-4o-mini. We selected three datasets of different types to evaluate the adaptability of the memory mechanisms across multiple scenarios. Table 3 reports the performance on three types of datasets using a single random seed. When both memory mechanisms are included and operate in coordination, the overall performance is substantially higher than in the other configurations, demonstrating the effectiveness and complementary benefits of the proposed memory design.

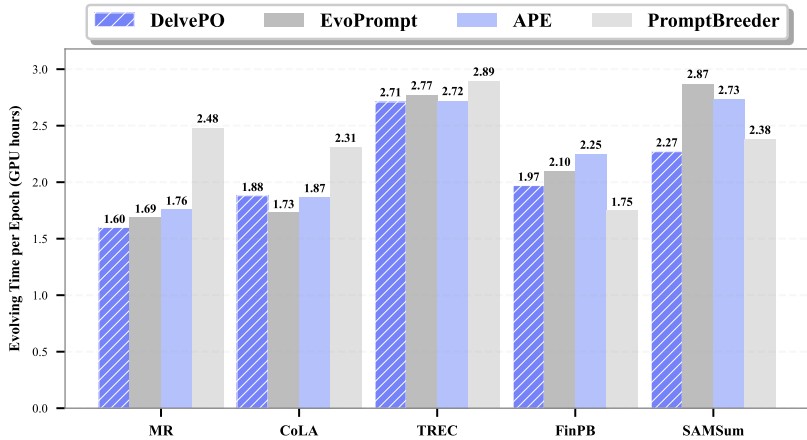

Figure 3: Average time-consuming (GPU hours) for one epoch of optimization on DeepSeek-R1-Distill-Llama-8B.

Table 3: Ablations of Memory Mechanism.

| Memory Modules | SAMSum | SQuAD | Causal Judgement |
|---|---|---|---|
| w/o Component Memory | 28.8 | 67.4 | 62.6 |
| w/o Prompt Memory | 29.4 | 67.9 | 61.8 |
| w/o both | 28.4 | 64.6 | 61.3 |
| **DelvePO** | **35.3** | **84.7** | **65.7** |

Furthermore, to investigate the impact of the number of component values for each component type on the overall performance of the initial population, we designed a sensitivity test examining how initial population performance varies with the number of component values at initialization. Using GPT-4o-mini, we generated initial populations for three different types of datasets under a single random seed and evaluated their performance on the corresponding test sets. The results in Table 4 show that in-

Table 4: Sensitivity test regarding the number of component values.

| # Value | SAMSum | SQuAD | SST-5 |
|---|---|---|---|
| 50 | 29.2 | 67.9 | 57.2 |
| 40 | 29.2 | 67.3 | 57.4 |
| 30 | 29.7 | 66.8 | 56.8 |
| 20 | 28.8 | 66.5 | 59.1 |
| 10 | **30.2** | **69.7** | **60.3** |

creasing the number of component values does not cause significant fluctuations in the initial population performance. This indicates that a relatively small number of component values is sufficient to obtain an initial population with stable and reasonable performance, and importantly, it rules out the concern that a larger number of components could lead to an overestimated initial population, which might otherwise suggest that further optimization is unnecessary.

To illustrate the stability of our method, we use the MR dataset as an example and report the average and best population performance over 10 epochs (Figure 5, Appendix D). As iterations increase, the performance population of DelvePO steadily improves, while baselines exhibit larger fluctuations, demonstrating its robustness. We also conducted a **case study** to help researchers quickly understand our framework, with details in Appendix I.

## 5    RELATED WORK

**Prompt Engineering**    Prompt engineering is a resource-efficient approach, focusing on elaborately designing expert-level prompts to steer LLMs generate desired solutions to various downstream tasks. In this part, we mainly focus on those works which use prompts to stimulate the internal abilities of LLMs. Least-to-Most (Zhou et al., 2023a), Decomposed Prompting (Khot et al., 2023) and PS&PS+ (Wang et al., 2023a) use prompts to leverage the decomposition ability of LLMs, breaking

down complex problems into simpler ones, enabling the model to perform better when dealing with complex problems. CoT (Wei et al., 2023), PoT (Chen et al., 2023), PS & PS+ (Wang et al., 2023a), Automate-CoT (Shum et al., 2023), ToT (Yao et al., 2023)and GoT (Besta et al., 2024) guide the model to utilize chain-of-thought in different ways through the design of prompts, stimulating the thinking ability of the model, thereby enhancing the model's reasoning ability. Also, Complexity-based Prompting (Fu et al., 2023) and DIV-SE (Naik et al., 2024) focus on the complexity and diversity of prompts, aiming to help the model think better. Rephrase and Respond (Deng et al., 2024), OPRO (Yang et al., 2024), and MIPRO (Opsahl-Ong et al., 2024) utilized the self-optimization capabilities of LLMs through methods such as input rewriting, iterative prompt optimization and structured program optimization, jointly demonstrating that LLMs can autonomously enhance the performance of task execution by dynamically improving prompts. TextGrad (Yuksekgonul et al., 2025) and SPO (Xiang et al., 2025) combine LLMs by orchestrating Standard Operation Pipelines (SOPs) in advance, and uses the evaluation ability of the model itself to guide the optimization of prompts. These methods effectively demonstrate that LLMs can be more proactive in utilizing their exploration abilities under the scientific guidance of predefined SOPs. Although the above works have elicited some abilities of LLMs to cope with complex problems, they cannot get rid of the problem that LLMs are sensitive to inputs, which results in the inconsistency of outputs' quality.

**Prompt Optimization** Given a downstream task, prompt optimization aims to improve the effectiveness of prompt, which typically involves an iterative process including initialization, execution, evaluation and selection. This part primarily focus on those works which leverage external technologies or exogenous intelligence sources to guide LLMs to perform prompt optimization. Using external knowledge to optimize prompt is very effective. Existing works generally referred to: 1) the way humans think (Wang et al., 2023c); 2) the idea of program synthesis (Zhou et al., 2023b); 3) external knowledge (Zhao et al., 2023) to optimize prompts which achieve good results. Formatting the structure of prompts can standardize the thinking process of LLMs, and to a certain extent improve their reasoning capability. LangGPT (Wang et al., 2024) presents a framework for prompt design, proving that scalable structures are important for prompts migration. Prompt template (He et al., 2024) delves into the impact of the format of the prompt template on solving problems, demonstrating the effectiveness of structured prompts in eliciting LLMs' capabilities. Furthermore, there are some efforts that introduce algorithms that have been widely proven to have good optimization capabilities to the optimization of prompts, including K-means (Zhang et al., 2022), KNN (Shi et al., 2022), reinforcement learning (Pryzant et al., 2023; Wang et al., 2023b), active learning (Diao et al., 2024), and evolutionary algorithm (Guo et al., 2025; Fernando et al., 2024).

In summary, although existing studies have mitigated the output stochasticity of LLMs, the efficiency of the optimization algorithm has still not been fully explored. These efforts generally tend to treat prompts as a whole unit to optimize, so the potential optimization space is very large. In addition, most previous researches combining optimizing algorithms (e.g., evolutionary algorithms) with LLMs, do not take full advantage of the experience generated before and after optimization, so that the optimization process is more stochastic, which tends to fall into local optima. Inspired by biological Loci and Alleles, this paper proposes a flexible framework for prompt optimization, which can effectively reduce the randomness of the optimization process and significantly improves the optimization speed. We hope our approach will provide possible improvements for subsequent PO methods, significantly lowering the learning barrier for non-AI experts to leverage LLMs.

## 6 CONCLUSION

We introduced DelvePO, a self-evolving framework for prompt optimization that decouples prompts into distinct components. With components, prompts can be modified by adding or removing content that may affect their performance, striking a good balance between exploration and exploitation of factors that affect task performance. DelvePO employs a co-evolutionary mechanism to iteratively refine the specifics of two sub-tasks and generate corresponding solutions. The evolved prompt, following systematic processing, is encoded into working memory to facilitate LLMs in deriving relevant insights, thereby provides directional guidance for generating task-specific prompts. Extensive experiments on different tasks demonstrate DelvePO consistently outperforms baselines, validating its effectiveness. As we anticipate the emergence of even more powerful LLMs that can deal with longer context, we firmly believe that more professional prompts will penetrate all walks of life, and DelvePO will help more users complete various complex tasks.

## ETHICS STATEMENT

This work studies prompt optimization techniques for language models (LLMs) to better elicit their capabilities in solving target tasks. The primary potential risks of this research are related to the misuse of LLMs, for example, generating misleading, harmful, or biased content.

In our experiments, we only use publicly available datasets and pre-trained LLMs, and no private or sensitive data were involved. Specific statements on LLM usage can be found in Appendix A. We emphasize that our methods are intended for research and benchmarking purposes, and we encourage responsible use to mitigate potential societal risks.

## REPRODICIBILITY STATEMENT

We are committed to ensuring the reproducibility of our work. To facilitate replication, we provide the following details:

**Computational Resources** The following describes the experimental environment, including detailed information on both hardware and software configurations.

- **Hardware**. All experiments were conducted on a computing node equipped with four NVIDIA Tesla V100-SXM2 GPUs (32GB memory each), an Intel Xeon Gold 6248 CPU @ 2.50GHz with 20 cores, and 226 GB of RAM.
- **Software**. The system runs Ubuntu 20.04.6 LTS with Linux kernel version 5.4.0. All models were implemented in Python 3.10.18 using PyTorch 2.0.0 with CUDA 11.7.

**Hyperparameter Details** In order to isolate the effect of our proposed method and ensure a fair comparison, we mainly followed the default configurations used in baseline methods and intentionally introduced no additional trainable parameters. Specifically, the detailed hyperparameter settings are given below.

- **Initial Population Size**. Following the setup of EvoPrompt, which uses both human-written and LLM-generated prompts, we adopted a similar strategy in spirit but tailored it to our fully automated framework. (1) We identify a fixed set of components through preliminary study mentioned at ref . (2) For each component, we use an LLM to generate 10 candidate values based on prompt templates. (3) We then randomly combine these values to create 10 initial prompts, which together form the initial population for the evolutionary process.
- **Temperature**. Since the stochasticity of LLM outputs is sensitive to temperature settings, we set the temperature to 0.5 to strike a balance between exploration and exploitation. This choice aligns with prior work such as EvoPrompt.
- **Sample Allocation**. For data splits, we followed the protocols of APE and EvoPrompt. Specifically, if the dataset has a predefined training/testing split, we used it as-is. For datasets without predefined splits, we randomly selected 100 examples as the test set and used the remaining examples for training.
- **Randomness Control**. To ensure reproducibility. Unless otherwise noted, we use 3 random seeds (5, 10 and 15) in the training phrase, and reported the results on the test set.

## LIMITATIONS

While our framework can adaptively design well-matched prompts for any LLM across diverse downstream tasks, several limitations remain. (1) Due to substantial computational costs, we cannot comprehensively evaluate all models and domains. Instead, we focused on widely used datasets to balance fairness and coverage. (2) Although we report monetary cost based on actual token usage, variations in token pricing across input and output types cannot be precisely captured by the API. Analysis indicates that most of the cost arises from including memory content as input tokens, while output token consumption remains relatively modest, particularly when "thinking mode" is disabled. Future work will explore prompt compression to further optimize resource use. (3) We evaluated only representative component values from each category due to resource constraints. Nevertheless, even with this limited set, our approach continues to outperforms or remains competitive with baselines, demonstrating its effectiveness and suggesting that its benefits will likely increase as LLMs support longer contexts.

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

## A  USE OF LLMs

Large Language Models (LLMs) were used in two ways in this work. First, LLMs served as base models in our experiments on prompt optimization, where we studied how different prompts can elicit their capabilities to solve target tasks. Second, LLMs were employed as auxiliary tools for minor writing support, such as grammar checking and phrasing improvements. Specific details about the LLMs used in our experiments can be found in Appendix B. No LLMs were used to generate substantive ideas, analyses, or content of the paper.

## B  DETAILS OF DATASETS AND LLMs USED

**Datasets** For fair comparison, we followed the datasets and evaluation metrics used in prior baselines whenever possible. Specifically, we include 4 classic NLP benchmarks (*MR*, *Subj*, *CoLA*, *SST-5*) and two widely used question-answering datasets (*SQuAD*, *TREC*) to validate basic capabilities; several domain-specific benchmarks to probe specialized performance, including *Financial Sentiment Evaluation* dataset (*FinFE*), *Financial PhraseBank* (*FinPB*), reasoning related dataset (*Casual Judgement*). Besides, one multi-domain datasets (*AG's News*) and one natural language generation dataset (*SAMSum*) are also used to assess overall robustness. To evaluate output quality beyond simple accuracy, we report ROUGE-Avg on *SAMSum* and the Matthews correlation coefficient (MCC) on *CoLA*. To balance computational cost while maximizing coverage, we selected datasets according to a "maximize capability diversity" principle — for example, in addition to the main experiments we ran Qwen2.5-7B-Instruct on *Subj*, *AG's News*, and *FinFE* to cover several of the categories above. Detailed results are presented in the experimental analysis section.

**LLMs** To demonstrate the adaptability of the proposed method for LLMs, we selected *DeepSeek-R1-Distill-Llama-8B* and *Qwen2.5-7B-Instruct* from open-source LLMs, as well as *GPT-4o-mini* from closed-source LLMs, as the base models for our experiments. The experiments on *DeepSeek-R1-Distill-Llama-8B* evaluate both the performance of the DeepSeek model itself and, to some extent, the capabilities of the underlying Llama architecture, which is primarily trained on English-language data. Experiments on *Qwen2.5-7B-Instruct* assess the framework's performance on a model predominantly trained on Chinese-language data, demonstrating applicability to non-English corpora. *GPT-4o-mini* was included because it is a widely used closed-source model in prior studies and allows cost-effective experimentation within our budget.

## C  ALGORITHM DETAILS

---
**Algorithm 1** An Overview of **DelvePO**

---
**Require:** A population of prompts $\boldsymbol{P}$, size of population $N$, task-related dataset $\boldsymbol{D}$, number of epochs $m$, number of iterations $n$, working memory $\boldsymbol{M} = \{\boldsymbol{M}_{\text{components}}, \boldsymbol{M}_{\text{prompts}}\}$
**Ensure:** Best prompt $p^*$
1: **Initialization:** $\boldsymbol{P} = \{p_1, p_2, \cdots, p_N\}$, $\boldsymbol{M}_{\text{prompts}} \leftarrow f_{sort}(\boldsymbol{P})$, $\boldsymbol{M}_{\text{components}} \leftarrow \emptyset$
2: **for** epoch = 1 to $m$ **do**
3:    $\boldsymbol{P}_{\text{evo}} \leftarrow \emptyset$
4:    **for** step = 1 to $n$ **do**
5:       **Selection:** $p \leftarrow f_{r.w.s.}(\boldsymbol{P})$
6:       **Task-Evolution:** $\mathcal{T}_{\text{evo}} \leftarrow \phi^{\mathcal{T}}(p, \boldsymbol{M}_{\text{components}} \mid \mathcal{T})$
7:       **Solution-Evolution:** $\mathcal{S}_{\text{evo}} \leftarrow \phi^{\mathcal{S}}(p, \boldsymbol{M}_{\text{prompts}} \mid \mathcal{T}_{\text{evo}})$
8:       **Evaluation:** $p' \leftarrow \phi^{\mathcal{LLM}}(\mathcal{S}_{\text{evo}})$, $s' \leftarrow f_{eval}(p', \boldsymbol{D})$
9:       **Memory-Evolution:** $\boldsymbol{M}_{\text{evo}} \leftarrow \phi^{\mathcal{M}}(\boldsymbol{M}, \langle p, p', s \geq s' \rangle)$
10:      $\boldsymbol{P}_{\text{evo}} \leftarrow \{\boldsymbol{P}_{\text{evo}}, p'\}$
11:   **end for**
12:   **Update:** $\boldsymbol{P} \leftarrow \text{Top-}N\{\boldsymbol{P}, \boldsymbol{P}_{\text{evo}}\}$
13: **end for**
14: **Return** the best prompt $p^*$: $p^* \leftarrow \underset{p \in \boldsymbol{P}}{\arg\max} \, f_{eval}(\phi^{\mathcal{LLM}}(p, \boldsymbol{D}))$

---

The sampling function used in our framework is roulette wheel selection, denoted as $f_{r.w.s.}(\cdot)$, which is commonly used in the evolution algorithm. $\phi^{\mathcal{T}}$, $\phi^{\mathcal{S}}$, $\phi^{\mathcal{M}}$ refer to the Task-Evolution, Solution-Evolution, Memory-Evolution methods, respectively. Similarly, $\mathcal{T}$, $\mathcal{S}$, and $\mathcal{M}$ mean the corresponding Task, Solution, Memory. Based on the components, we designed a task-agnostic template described in Figure 4, through which any kind of LLMs can construct an initial content set of components based on a simple description of the target task input by the user.

---

================== **Task-agnostic Template for Component** ==================

Hi there, I have a task to do which can be described as ***Downstream Task Related Information***. Now I want you to give me N contents related to Component. [OPTIONAL Example]. Please list your answers in the following format: ['content 1', 'content 2',...]

-------------------------- **Downstream Task (Causal judgement) for role** --------------------------

\<Query\>: Hi there, I have a task to do which can be described as "answer questions about causal attribution". Now I want you to give me 10 related roles who are expertise in these questions. For example, 'Casusal Analysis Experts', etc. Please list your answers in the following format: ["content 1", "content 2",...]

\<Response\>: ["Cognitive Scientist", "Social Psychologist", "Computational Linguist", "AI Ethicist", "Behavioral Economist", "Decision Theorist", "Philosophy of Mind Researcher", "Causal Inference Data Scientist", "Educational Psychologist", "Human-Computer Interaction Specialist"]

---

Figure 4: Task-agnostic template for generating component values corresponding to the given component types. *The following part of the figure is the prompt to generate content for Component "role" using the casual judgement task as an example.*

# D ADDITIONAL EXPERIMENTS

Table 5: The results on different downstream tasks for Qwen2.5-7B-Instruct.

| Method | Classical NLP | | | Question-Answering | Domain-specific | Multi-domain | Avg. |
|---|---|---|---|---|---|---|---|
| | **Subj** | **SST-5** | **CoLA** | **TREC** | **FinFE** | **AG's News** | |
| APE | $69.00_{(3.06)}$ | $47.00_{(1.10)}$ | $79.05_{(1.73)}$ | $43.40_{(1.14)}$ | $64.30_{(2.70)}$ | $83.43_{(1.90)}$ | 64.38 |
| EvoPrompt | $\underline{77.03}_{(4.74)}$ | $\underline{57.67}_{(1.19)}$ | $\underline{79.69}_{(1.42)}$ | $\underline{67.55}_{(2.08)}$ | $\underline{64.67}_{(1.22)}$ | $\underline{85.73}_{(1.29)}$ | $\underline{72.06}$ |
| **DelvePO** | $\mathbf{80.07}_{(0.65)}$ | $\mathbf{60.00}_{(1.69)}$ | $\mathbf{81.40}_{(1.07)}$ | $\mathbf{70.77}_{(1.74)}$ | $\mathbf{69.97}_{(0.87)}$ | $\mathbf{89.27}_{(0.97)}$ | **75.25** |

Table 6: Average monetary cost (USD) for one epoch of optimization on GPT-4o-mini.

| Methods | Subj | CoLA | FinPB | AG's News |
|---|---|---|---|---|
| Promptbreeder | 1.17 | 1.31 | 0.97 | 1.52 |
| APE | 0.57 | 0.56 | 0.61 | 0.79 |
| EvoPrompt | 0.83 | 0.64 | 0.74 | 1.23 |
| DelvePO | 1.27 | 1.08 | 1.30 | 1.10 |

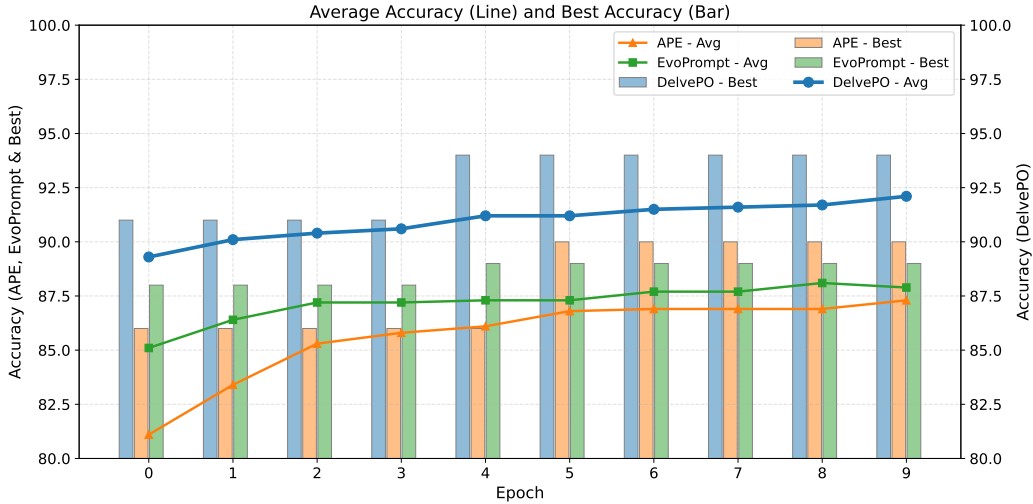

Figure 5: Robustness of DelvePO as the number of epochs increases (Take the dataset MR as an example).

## E   DETAILED INFORMATION ABOUT COMPONENTS

To ensure that the types of components are as comprehensive and representative as possible, we first surveyed a broad set of related literature (Yuksekgonul et al., 2025; He et al., 2024; Feng et al., 2024; Opsahl-Ong et al., 2024; Diao et al., 2024; Wang et al., 2024; 2023b) and extracted a variety of factors that have been shown to influence the performance of prompts, forming our component pool. We then categorized all components in the pool based on the semantics implied in their original sources, which resulted in five categories: "Role and Expertise", "Task Content", "Constraints and Norms", "Process and Behavior" and "Context and Examples". From each category, we selected the most representative component as our predefined component types. The complete component pool and its categorization are provided in Table 7.

Despite this extensive literature review, we acknowledge that some important aspects may remain uncovered. This observation motivated our design: as more non-AI experts begin to use LLMs, domain specialists should be able to adaptively define new components through our mechanism, thereby supporting both effective task performance and improved interpretability. It is worth noting that for each component type, we can add a "null" option when generating its values, allowing the presence or absence of the component to be controlled and makes the optimized prompts more flexible.

Table 7: The categories and types of components in the component pool

| Categories | Related Items |
|---|---|
| Role and Expertise | Role; Role description; Scenario; Domain knowledge; Term Clarification |
| Task Content | Task description; Instruction; Goal |
| Constraints and Norms | Output format; Constraints; Principle; Style; Length; Tone; Priority & Emphasis; Exception handling; Target audience |
| Process and Behavior | Workflow; CoT; Action; Skill; Suggestions; Initialization |
| Context and Examples | Examples; Reference prompt; Attachment |

## F   TEMPLATE FOR INJECTION & PROMPTS FOR EVALUATION ON LLMS

**Template_For_Injection** General Form =

··· `<component1>`{content1}`</component1>`. Given the Input, ··· `<component2>`{content2}`</component2>` ···

**Template_For_Injection** AG's News =

You are a `<role>`{role}`</role>`. Given the News, your task is to `<task_description>`{task_description}`</task_description>`.

**Template_For_Injection** Simplification =

You are a `<role>`{role}`</role>`. Given the English Sentence, your task is to `<task_description>`{task_description}`</task_description>`.

Figure 6: Template for initializing prompt populations. *It is also used in the construction of Prompts Memory, that is, injecting discrete components into the template to obtain a continuous form prompt. The above shows the general form, while the two below provide illustrative examples.*

**Prompt_For_LLM** General Form =

<INSTRUCTION>: · · · {**content1**}. Given the Input, · · · {**content2**} · · ·

<Input>: {**input**}

<OUTPUT FORMAT>: Output the final result starting with the tag <res> and ending with the tag </res>. [OPTIONAL REQUIREMENTS]

**Prompt_For_LLM** AG's News =

<INSTRUCTION>: You are a {**role**}. Given the News, your task is to {**task_description**}.

<News>: {**input**}

<OUTPUT FORMAT>: Output the final result starting with the tag <res> and ending with the tag </res>. The final result must come from the following: [World, Sports, Business, Tech].

**Prompt_For_LLM** Simplification =

<INSTRUCTION>: You are a {**role**}. Given the English Sentence, your task is to {**task_description**}.

<English Sentence>: {**input**}

<OUTPUT FORMAT>: Output the final result starting with the tag <res> and ending with the tag </res>.

Figure 7: Complete prompt template for LLMs (including three parts: instruction, input, and output). *Here we also display two practical prompts for AG's News and Simplification Tasks*.

## G  THE DETAILED PROMPTS OF TASK-EVOLUTION

---

Please follow the instructions step-by-step to get **final result**.

`Step 1` Conclude **Insights** from the provided `Memory Components`, which consists of multiple elements. Each element contains two lists: the first contains several markup pairs in the format `<component>`content`</component>`. For example, in the pair `<role>`role_description`</role>`, the content ("role_description") describes the component ("role"). All markup pairs follow this structure. By default, the first list in each element is considered to perform better than the second.
`Memory Components`: {$M_{\text{components}}$}

`Step 2` Based on the **Insights** from Step 1 and the `Current Prompt`, select one or more component(s) from `Component Set` that could potentially improve performance to form **final result**. Separate the final result with a special token '|' and ensure that each of final result is unique and appears only once. The final result must start with the tag `<res>` and end with the tag `</res>`. For example, the final result must follow the format: `<res>`component1|...`</res>`.
`Current Prompt`: { $p$ }
`Component Set`: {components}

---

Figure 8: The prompts for sub-task I

---

Please follow the instructions step-by-step to get **final result**.

`Step 1` Conclude **Insights** from the provided `Memory Components`, which consists of multiple elements. Each element contains two lists: the first contains several markup pairs in the format `<component>`content`</component>`. For example, in the pair `<role>`role_description`</role>`, the content ("role_description") describes the component("role"). All markup pairs follow this structure. By default, the first list in each element is considered to perform better than the second.
`Memory Components`: {$M_{\text{components}}$}

`Step 2` Given a list named `Old Values`, where each element contains a pair of contents, use the **Insights** from Step 1 to select one content from each pair in original order. The **final result** must start with the tag `<res>` and end with the tag `</res>`. For example, the final results must follow the format: `<res>`content1|...`</res>`.
`Old Values`: {old_values}

---

Figure 9: The prompts for sub-task II

## H  THE DETAILED PROMPTS OF SOLUTION-EVOLUTION

Please follow the instructions step-by-step to get **final result**.

Step 1  Conclude the **Insights** from the Memory Prompts, which consists of multiple items. Each item includes two parts: the first part contains several markup pairs in the format <component>content</component>. For example, in the pair <role>role_description</role>, the content ("role_description") describes the component ("role"). Other markup pairs follow this same structure. The second part of each item represents its corresponding performance. The entire Memory Prompts is sorted in descending order based on performance.

Memory Prompts : $\{M_{\text{prompts}}^{\text{discrete}}\}$

Step 2  Given a list named Old Values , use the **Insights** from Step 1 to generate a new mutated content for each content to form a new list, i.e. **final result**, referring to Description, adhering to Rules below.

Description:

- In Old Values , each element is a markup pair like <component>content</component> containing content that needs to mutate.

Rules:

1. Mutation Requirements:
   - For each element like <component>content</component>, generate a **new one content** that:
     - If the component is <role>, the new content must be a **noun phrase** describing a person.
     - If the component is <task_description>, the new content must be a **verb phrase** describing a task.
     - Is **distinct** from the original content.
     - Preserves lexical identity (noun/verb phrase) matching the component.
     - If the original content had the **highest score**, the new content must prioritize **improved performance potential** (e.g., higher efficiency, enhanced properties).
     - Otherwise, the new content may be derived from those contents linked to its corresponding component in the Memory Prompts (optional but allowed).
2. Output Format:
   - Start with <res> and end with </res>.
   - Separate mutated contents **strictly** with '|' (no extra characters).
   - Never include original contents in the output.

Old Values : {old_values}

Figure 10: The prompts for Sub-solution I - Prompts Memory in discrete form

Please follow the instructions step-by-step to get **final result**.

Step 1 Conclude the **Insights** from the Memory Prompts, which contains multiple items. Each item has two parts: a sentence enclosed in <prompt> and </prompt>, and its corresponding performance score. The sentence includes markup pairs in the format <component>content</component>, where the content describes the component. For example, <role>role_description</role> indicates that "role_description" explains the "role" component. All items are sorted in descending order by performance.

Memory Prompts : $\{M_{\text{prompts}}^{\text{continuous}}\}$

Step 2 Based on the Current Prompt and **Insights** from Step 1, generate a new mutated content for each markup pair whose component matches those listed in Mutate Factors to form the **final result**, referring to Description, adhering to Rules below.

Description:

- In Current Prompt, markup pair like <component>content</component> contains content that needs to mutate.
- In Mutate Factors, each element is a component appeared in Current Prompt.

Rules:

1. Mutation Requirements:
   - For each markup pair like <component>content</component>, if the component in Mutate Factors, generate a **new one content** that:
     - If the component is <role>, the new content must be a **noun phrase** describing a person.
     - If the component is <task_description>, the new content must be a **verb phrase** describing a task.
     - Is **distinct** from the original content.
     - Preserves lexical identity (noun/verb phrase) matching the component.
     - If the original content had the **highest score**, prioritize generating contents with **improved performance potential** (e.g., higher efficiency, enhanced properties).
     - Otherwise, the new content may derive from those contents linked to its component in the Memory Prompts (optional but allowed).

2. Output Format:
   - Start with <prompt> and end with </prompt>.
   - **Only mutate contents within markup pairs specified in** Mutate Factors.
   - Preserve all other values outside markup pairs.
   - Replace original contents with mutated ones directly within their components.

Current Prompt : {prompt}
Mutate Factors : {mutate_factors}

Figure 11: The prompts for Sub-solution I - Prompts Memory in continuous form

Please follow the instructions step-by-step to get **final result**.

Step 1 Conclude the **Insights** from the Memory Prompts, which consists of multiple items. Each item includes two parts: the first part contains several markup pairs in the format `<component>`content`</component>`. For example, in the pair `<role>`role_description`</role>`, the content ("role_description") describes the component ("role"). Other markup pairs follow this same structure. The second part of each item represents its corresponding performance. The entire Memory Prompts is sorted in descending order based on performance.

Memory Prompts : $\{M_{\text{prompts}}^{\text{discrete}}\}$

Step 2 Given a list named Old Values, where each element contains a pair of contents, use the **Insights** from Step 1 to generate a new mutated content for each pair to form a new list, i.e. **final result**, referring to Description, adhering to Rules below.

Old Values : {old_values}

Description:

- In Old Values, each element contains a pair of contents like **[a, b]**.

Rules:

1. Mutation Requirements:
    - For each pair of contents like **[a, b]**, generate a **new one content** that:
        - If **a** and **b** are enclosed with `<role>` & `</role>`, the new content must be a noun phrase used to describe a person.
        - If **a** and **b** are enclosed with `<task_description>` & `</task_description>`, the new content must be a verb phrase used to describe a task.
        - Is **distinct** from both **a** and **b**.
        - Preserve corresponding lexical identity.
        - If the original pair has the **highest score**, prioritize generating contents with **improved performance potential** (e.g., higher efficiency, enhanced properties).
        - Otherwise, derive the new content from those contents linked to its component in the Memory Prompts (optional but allowed).
2. Output Format:
    - Start with `<res>` and end with `</res>`.
    - Separate mutated contents **strictly** with '|' (no extra characters).
    - Never include original pairs in the output.

Figure 12: The prompts for Sub-solution II - Prompts Memory in discrete form

Please follow the instructions step-by-step to get **final result**.

Step 1 Conclude the **Insights** from the Memory Prompts, which contains multiple items. Each item has two parts: a sentence enclosed in \<prompt\> and \</prompt\>, and its corresponding performance score. The sentence includes markup pairs in the format \<component\>content\</component\>, where the content describes the component. For example, \<role\>role_description\</role\> indicates that "role_description" explains the "role" component. All items are sorted in descending order by performance.

Memory Prompts : $\{M_{\text{prompts}}^{\text{continuous}}\}$

Step 2 Based on the Prompt 1 and **Insights** from Step 1, generate a new mutated content for each markup pair whose component matches those listed in Mutate Factors to form the Prompt 2, referring to Description, adhering to Rules below.

Description:

- In Prompt 1, markup pair like \<component\>content\</component\> contains content that needs to mutate.

- In Mutate Factors, each element is a content appeared in Prompt 1.

Rules:

1. Mutation Requirements:

   - For each markup pair like \<component\>content\</component\>, if the component in Mutate Factors, Generate a **new one content** that:

     - If the component is \<role\>, the new content must be a **noun phrase** describing a person.

     - If the component is \<task_description\>, the new content must be a **verb phrase** describing a task.

     - Is **distinct** from the original content.

     - Preserves lexical identity (noun/verb phrase) matching the component.

     - If the original content had the **highest score**, prioritize generating contents with **improved performance potential** (e.g., higher efficiency, enhanced properties).

     - Otherwise, the new content may derive from those contents linked to its component in the Memory Prompts (optional but allowed).

2. Output Format:

   - Start with \<prompt\> and end with \</prompt\>.

   - **Only mutate contents within markup pairs specified in** Mutate Factors.

   - Preserve all other values outside markup pairs.

   - Replace original contents with mutated ones directly within their components.

Prompt 1 : {prompt1}

Mutate Factors : {mutate_factors}

Step 3 Based on the Prompt 3 and **Insights** from Step 1, generate a new mutated content for each markup pair whose component matches those listed in Mutate Factors to form the Prompt 4, referring to Description, adhering to Rules below.

Description:

- In Prompt 3, markup pair like \<component\>content\</component\> contains content that needs to mutate.

Figure 13: The prompts for Sub-solution II - Prompts Memory in continuous form

- In Mutate Factors, each element is a content appeared in Prompt 3.

Rules:

1. Mutation Requirements:

   - For each markup pair like <component>content</component>, if the component in Mutate Factors, Generate a **new one content** that:

     - If the component is <role>, the new content must be a **noun phrase** describing a person.

     - If the component is <task_description>, the new content must be a **verb phrase** describing a task.

     - Is **distinct** from the original content.

     - Preserves lexical identity (noun/verb phrase) matching the component.

     - If the original content had the **highest score**, prioritize generating contents with **improved performance potential** (e.g., higher efficiency, enhanced properties).

     - Otherwise, the new content may derive from those contents linked to its component in the Memory Prompts (optional but allowed).

2. Output Format:

   - Start with <prompt> and end with </prompt>.

   - **Only mutate contents within markup pairs specified in Mutate Factors**.

   - Preserve all other values outside markup pairs.

   - Replace original contents with mutated ones directly within their components.

Prompt 3 : {prompt3}

Mutate Factors : {mutate_factors}

**Step 4** Generate **final result** by selecting contents from pairs in Prompt 2 and Prompt 4 under identical markup components, referring to Description, adhering to Rules below.

Description:

- Pairs from Prompt 2 and Prompt 4 have identical components (e.g., <role>, <task_description>).

Rules:

1. Selection Criteria:

   - For each tagged pair (e.g., <role>a</role> and <role>b</role>):

     - Use **Insights** from Step 1 to **select one content** (a or b) that has **higher performance improvement potential** (e.g., clarity, specificity, alignment with goals).

     - If the component is <role>, the new content must be a **noun phrase** describing a person.

     - If the component is <task_description>, the new content must be a **verb phrase** describing a task.

     - Preserve the lexical identity of the component.

     - Never modify text **outside** markup pairs.

2. Output Format:

   - Start with <prompt> and end with </prompt>.

   - Retain the structure of Prompt 3 but replace tagged pairs with the selected contents.

   - If multiple tagged pairs exist, update all while maintaining non-tagged values verbatim.

Figure 14: The prompts for Sub-solution II - Prompts Memory in continuous form (extended from Figure 13)

# I CASE STUDY DETAILS

To quickly verify the generalizability of our framework, we conducted multi-turn dialogues with DeepSeek Chat via the web interface provided by DeepSeek (DeepSeek Chat, 2025).

Throughout the process, we take simplification task (Zhang et al., 2023) as the example, which allows for easy observation and interpretation of the outputs, and randomly set 4 components. The whole process can be find in Appendix I. For Task-Evolution, we provide two input information (see Figure 15, 17) for the prompt of two sub tasks (see Figure 8, 9). And the corresponding outputs are shown in Figure 16, 18. From the final results, we can derive that under the guidance of direction (i.e., Memory Components), The LLMs could find reasonable direction for evolutionary operator.

Accordingly, for Solution-Evolution, we provide four input information (see Figure 19, 21, 23, 25) for four kinds of scenarios. And the corresponding responses are shown in Figure 20, 22, 24, 26, respectively. Based on the observation from the responses, we also find that the procedure designed in this paper can accurately guide the model to deduce corresponding insights and further output reasonable results based on the insights.

By the way, this case study provides new users with a quick understanding of our framework. It can also serve as a practical guide, allowing anyone to construct a prototype using the system prompts provided in the case study without writing any code, which can help users optimize domain-specific prompts. We sincerely invite researchers to try it out and share their valuable feedback for further improvement.

```
Step 1  Memory Components:

[
  [
    [<action>Clarify Sentence Meaning</action>, <requirements>Adapt to
Audience</requirements>],
    [<action>Reduce Linguistic Complexity</action>, <requirements>Improve
Structure</requirements>]
  ],
  [
    [<role>Readability Specialist</role>],
    [<role>Sentence Simplifier</role>]
  ]
]

Step 2  Current Prompt:

[
    <role>Sentence Simplifier</role>,
    <task_description>Rewrite the provided sentences in a simpler
form</task_description>,
    <action>Condense Sentences</action>,
    <requirements>Adapt to Audience</requirements>
]

Component Set:

[role, task_description, action, requirements]
```

Figure 15: The input for sub-task I

Step 1: Conclude **Insights** from the **Memory Components**

1. In the first tuple, `<action>Clarify Sentence Meaning</action>` is preferred over `<action>Reduce Linguistic Complexity</action>`, and `<requirements>Adapt to Audience</requirements>` is preferred over `<requirements>Improve Structure</requirements>`.

2. In the second tuple, `<role>Readability Specialist</role>` is preferred over `<role>Sentence Simplifier</role>`.

**Insights**:

- **Role**: "Readability Specialist" outperforms "Sentence Simplifier."
- **Action**: "Clarify Sentence Meaning" outperforms "Reduce Linguistic Complexity" (and by extension, other actions like "Condense Sentences").
- **Requirements**: "Adapt to Audience" is already the preferred choice.

Step 2: Select **Components** to improve performance based on **Current Prompt** and **Insights**

- The **Current Prompt** uses `<role>Sentence Simplifier</role>`, but the **Insights** suggest `<role>Readability Specialist</role>` is better.
- The **Current Prompt** uses `<action>Condense Sentences</action>`, but the **Insights** suggest `<action>Clarify Sentence Meaning</action>` is better.
- The **Current Prompt** already uses the preferred `<requirements>Adapt to Audience</requirements>`, so no change is needed here.

**final result**: `<res>role|action</res>`

Figure 16: The responses for sub-task I

Step 1 **Memory Components**:

```
[
    [
        [<action>Clarify Sentence Meaning</action>, <requirements>Adapt to
Audience</requirements>],                    [<action>Reduce Linguistic
Complexity</action>, <requirements>Improve Structure</requirements>]
    ],
    [
        [<role>Readability Specialist</role>],
        [<role>Sentence Simplifier</role>]
    ],
    [
        [<action>Optimize Readability</action>, <task_description>Reduce the
complexity of the provided sentences</task_description>],
        [<action>Reword for Better Clarity</action>, <task_description>Reword the
provided sentences for better clarity</task_description>]
    ]
]
```

Step 2 **Old Values**:

```
[
    [<role>Readability Specialist</role>, <role>Paraphrasing Specialist</role>],
    [<action>Simplify Sentence Structure</action>, <action>Break Down
Complexity</action>]
]
```

Figure 17: The input for sub-task II

**Step 1: Conclude Insights from Memory Components**

From the Memory Components:

1. **First Element:**
   - The pair `<action>Clarify Sentence Meaning</action>` with `<requirements>Adapt to Audience</requirements>` is preferred over `<action>Reduce Linguistic Complexity</action>` with `<requirements>Improve Structure</requirements>`.
   - **Insight:** Prioritizing audience adaptation and meaning clarification over general structural improvement and complexity reduction.

2. **Second Element:**
   - `<role>Readability Specialist</role>` is preferred over `<role>Sentence Simplifier</role>`.
   - **Insight:** "Readability Specialist" is considered a more effective or broader role than "Sentence Simplifier".

3. **Third Element:**
   - `<action>Optimize Readability</action>` is preferred over `<action>Reword for Better Clarity</action>`.
   - **Insight:** Optimizing readability takes precedence over merely rewording for clarity.

**Step 2: Apply Insights to Select from Old Values**

Old Values:

1. `[<role>Readability Specialist</role>, <role>Paraphrasing Specialist</role>]`
   - Based on Insight 2, "Readability Specialist" is already preferred, and since "Paraphrasing" is similar to "Sentence Simplifier", "Readability Specialist" aligns better with the insight.
2. `[<action>Simplify Sentence Structure</action>, <action>Break Down Complexity</action>]`
   - Based on Insight 1 and 3:
     - "Simplify Sentence Structure" relates more directly to optimizing readability and clarifying sentence meaning.
     - "Break Down Complexity" could be interpreted similarly to "Reduce Linguistic Complexity", which was not preferred.
   - So, choose "Simplify Sentence Structure".

**Final Output (final results):**

```
<res>
  <role>Readability Specialist</role>
  <action>Simplify Sentence Structure</action>
</res>
```

Figure 18: The responses for sub-task II

```
Step 1  Memory Promptsdiscrete:

[
    [
        [<role>Sentence Simplifier</role>, <task_description>Clarify the meaning of
    the given sentences by simplifying them</task_description>, <action>Optimize
    Readability</action>, <requirements>Improve Structure</requirements>],
        25.0
    ]
    [
        [<role>Readability Specialist</role>, <task_description>Reduce the complexity
    of the provided sentences</task_description>, <action>Clarify Sentence
    Meaning</action>, <requirements>Simplify Vocabulary</requirements>],
        21.0
    ]
    [
        [<role>Grammar & Clarity Editor</role>, <task_description>Modify the given
    sentences to be more straightforward</task_description>, <action>Reduce Linguistic
    Complexity</action>, <requirements>Adapt to Audience</requirements>],
        16.0
    ]
    [
        [<role>AI Sentence Simplifier</role>, <task_description>Reword the provided
    sentences for better clarity</task_description>, <action>Simplify Sentence
    Structure</action>, <requirements>Ensure Conciseness</requirements>],
        14.0
    ]
]

Step 2  Old Values:

[
    <role>Sentence Simplifier</role>,
    <action>Reduce Linguistic Complexity</action>
]
```

Figure 19: The input for Sub-solution I - Prompts Memory in discrete form

Based on the Memory Item List and the Old Values provided, here's how we can generate the Mutated Values:

1. For `<role>Sentence Simplifier</role>` :
   - This role already has the highest performance score (25.0), so we need to create a new value that's different but still relevant.
   - Possible mutation: "Clarity Optimizer" (derived from the action "Optimize Readability" in the top-performing item)

2. For `<action>Reduce Linguistic Complexity</action>` :
   - This action comes from an item with performance score 16.0 (not the highest), so we can derive a new value from higher-performing items.
   - The top-performing item has action "Optimize Readability" (score 25.0)
   - The second-highest has action "Clarify Sentence Meaning" (score 21.0)

Mutated Values:

```
<res>
    <role>Clarity Optimizer</role>,
    <action>Optimize Readability</action>
</res>
```

Figure 20: The responses for Sub-solution I - Prompts Memory in discrete form

Step 1 **Memory Prompts**[continuous]:

```
[
    [
        <prompt>You are a <role>Sentence Simplifier</role>. Given the English
Sentence, your task is to <task_description>Clarify the meaning of the given sentences
by simplifying them</task_description>. To accomplish this, you need to
<action>Optimize Readability</action>. Please make sure to <requirements>Improve
Structure</requirements> throughout the process.</prompt>,
        25.0
    ]
    [
        <prompt>You are a <role>Readability Specialist</role>. Given the English
Sentence, your task is to <task_description>Reduce the complexity of the provided
sentences</task_description>. To accomplish this, you need to <action>Clarify Sentence
Meaning</action>. Please make sure to <requirements>Simplify Vocabulary</requirements>
throughout the process.</prompt>,
        21.0
    ]
    [
        <prompt>You are a <role>Grammar & Clarity Editor</role>. Given the English
Sentence, your task is to <task_description>Modify the given sentences to be more
straightforward</task_description>. To accomplish this, you need to <action>Reduce
Linguistic Complexity</action>. Please make sure to <requirements>Adapt to
Audience</requirements> throughout the process.</prompt>,
        16.0
    ]
    [
        <prompt>You are a <role>Technical Content Simplifier</role>. Given the English
Sentence, your task is to <task_description>Reword the provided sentences for better
clarity</task_description>. To accomplish this, you need to <action>Simplify Sentence
Structure</action>. Please make sure to <requirements>Ensure
Conciseness</requirements> throughout the process.</prompt>,
        14.0
    ]
]
```

Step 2 **Current Prompt**:

```
<prompt>
    You are a <role>Sentence Simplifier</role>. Given the English Sentence, your task
is to <task_description>Rewrite the provided sentences in a simpler
form</task_description>. To accomplish this, you need to <action>Condense
Sentences</action>. Please make sure to <requirements>Adapt to Audience</requirements>
throughout the process.
</prompt>
```

Step 2 **Mutate Factors**:

```
<res> role | requirements </res>
```

Figure 21: The input for Sub-solution I - Prompts Memory in continuous form

Here is the **Final Prompt** with mutated values for the specified tags ( role and requirements ), while keeping the rest unchanged:

```
<prompt>
    You are a <role>Clarity Enhancer</role>. Given the English Sentence, your task is
to <task_description>Rewrite the provided sentences in a simpler
form</task_description>. To accomplish this, you need to <action>Condense
Sentences</action>. Please make sure to <requirements>Improve Structure</requirements>
throughout the process.
</prompt>
```

## Mutation Justification:

1. **<role>**
   - Original: *"Sentence Simplifier"*
   - Mutated: *"Clarity Enhancer"* (Inspired by high-performing roles like *"Readability Specialist"* and *"Grammar & Clarity Editor"*, but distinct.)

2. **<requirements>**
   - Original: *"Adapt to Audience"*
   - Mutated: *"Improve Structure"* (Drawn from the top-performing prompt's <requirements> , ensuring relevance while avoiding repetition.)

All other tags ( task_description , action ) remain **unchanged** as they were not in the Mutate Factors. The new values align with high-scoring patterns while being unique.

Figure 22: The responses for Sub-solution I - Prompts Memory in continuous form

```
Step 1  Memory Prompts^discrete:

[
    [
        [<role>Sentence Simplifier</role>, <task_description>Clarify the meaning of
the given sentences by simplifying them</task_description>, <action>Optimize
Readability</action>, <requirements>Improve Structure</requirements>],
        25.0
    ]
    [
        [<role>Readability Specialist</role>, <task_description>Reduce the complexity
of the provided sentences</task_description>, <action>Clarify Sentence
Meaning</action>, <requirements>Simplify Vocabulary</requirements>],
        21.0
    ]
    [
        [<role>Grammar & Clarity Editor</role>, <task_description>Modify the given
sentences to be more straightforward</task_description>, <action>Reduce Linguistic
Complexity</action>, <requirements>Adapt to Audience</requirements>],
        16.0
    ]
    [
        [<role>AI Sentence Simplifier</role>, <task_description>Reword the provided
sentences for better clarity</task_description>, <action>Simplify Sentence
Structure</action>, <requirements>Ensure Conciseness</requirements>],
        14.0
    ]
]

Step 2  Old Values:

[
    [<role>Sentence Simplifier</role>, <role>Paraphrasing Specialist</role>],
    [<action>Reduce Linguistic Complexity</action>, <action>Break Down
Complexity</action>]
]
```

Figure 23: The input for Sub-solution II - Prompts Memory in discrete form

**Step 1: Conclude the Insights from the Memory Prompts**

From the **Memory Prompts**, we can observe the following insights:

1. **Roles**: The roles are centered around simplifying, clarifying, or modifying sentences to improve readability and reduce complexity. Examples include "Sentence Simplifier," "Readability Specialist," "Grammar & Clarity Editor," and "AI Sentence Simplifier."

2. **Task Descriptions**: The tasks involve simplifying, rewording, or modifying sentences for clarity and straightforwardness. Examples include "Clarify the meaning of the given sentences by simplifying them" and "Reduce the complexity of the provided sentences."

3. **Actions**: The actions focus on optimizing readability, clarifying meaning, reducing linguistic complexity, and simplifying sentence structure. Examples include "Optimize Readability" and "Clarify Sentence Meaning."

4. **Requirements**: The requirements emphasize improving structure, simplifying vocabulary, adapting to the audience, and ensuring conciseness. Examples include "Improve Structure" and "Simplify Vocabulary."

The list is sorted in descending order of performance, with "Sentence Simplifier" being the highest-performing role.

**Step 2: Mutate a new value from each pair in Old Values**

**Old Value Pair 1**:

```
[<role>Sentence Simplifier</role>, <role>Paraphrasing Specialist</role>]
```

- **Sentence Simplifier**: From the Memory Prompts, this role involves simplifying sentences for clarity.
- **Paraphrasing Specialist**: Not explicitly in the Memory Prompts, but likely involves rewording sentences while preserving meaning.
- **Mutation**: Combine the ideas of simplifying and rewording to create a new role that emphasizes clarity and adaptation. For example, "Clarity Adaptation Editor" or "Simplification & Paraphrasing Expert."
  - Chosen mutation: **"Clarity Adaptation Editor"**

**Old Value Pair 2**:

```
[<action>Reduce Linguistic Complexity</action>, <action>Break Down
Complexity</action>]
```

- **Reduce Linguistic Complexity**: From the Memory Prompts, this involves making language less complex.
- **Break Down Complexity**: Similar, but more focused on decomposing complex ideas.
- **Mutation**: Combine the ideas of reducing and breaking down complexity into a more general or refined action. For example, "Streamline Language" or "Simplify and Deconstruct."
  - Chosen mutation: **"Streamline Language"**

**Final Mutated Values:**

```
<res>
    [<role>Clarity Adaptation Editor</role>, <action>Streamline Language</action>]
</res>
```

Figure 24: The responses for Sub-solution II - Prompts Memory in discrete form

**Step 1** **Memory Prompts**continuous:

```
[
    [
        <prompt>You are a <role>Sentence Simplifier</role>. Given the English Sentence,
your task is to <task_description>Clarify the meaning of the given sentences by
simplifying them</task_description>. To accomplish this, you need to <action>Optimize
Readability</action>. Please make sure to <requirements>Improve Structure</requirements>
throughout the process.</prompt>,
        25.0
    ]
    [
        <prompt>You are a <role>Readability Specialist</role>. Given the English
Sentence, your task is to <task_description>Reduce the complexity of the provided
sentences</task_description>. To accomplish this, you need to <action>Clarify Sentence
Meaning</action>. Please make sure to <requirements>Simplify Vocabulary</requirements>
throughout the process.</prompt>,
        21.0
    ]
    [
        <prompt>You are a <role>Grammar & Clarity Editor</role>. Given the English
Sentence, your task is to <task_description>Modify the given sentences to be more
straightforward</task_description>. To accomplish this, you need to <action>Reduce
Linguistic Complexity</action>. Please make sure to <requirements>Adapt to
Audience</requirements> throughout the process.</prompt>,
        16.0
    ]
    [
        <prompt>You are a <role>Technical Content Simplifier</role>. Given the English
Sentence, your task is to <task_description>Reword the provided sentences for better
clarity</task_description>. To accomplish this, you need to <action>Simplify Sentence
Structure</action>. Please make sure to <requirements>Ensure Conciseness</requirements>
throughout the process.</prompt>,
        14.0
    ]
]
```

**Step 2** **Prompt 1**:

```
<prompt>You are a <role>Sentence Simplifier</role>. Given the English Sentence, your task
is to <task_description>Rewrite the provided sentences in a simpler
form</task_description>. To accomplish this, you need to <action>Condense
Sentences</action>. Please make sure to <requirements>Adapt to Audience</requirements>
throughout the process.
</prompt>
```

**Mutate Factors**:

```
<res> role | requirements </res>
```

**Step 3** **Prompt 3**:

```
<prompt>You are a <role>Linguistic Analyst</role>. Given the English Sentence, your task
is to <task_description>Rewrite the provided sentences in a simpler
form</task_description>. To accomplish this, you need to <action>Condense
Sentences</action>. Please make sure to <requirements>Preserve Meaning</requirements>
throughout the process.
</prompt>
```

**Mutate Factors**:

```
<res> role | requirements </res>
```

Figure 25: The input for Sub-solution II - Prompts Memory in continuous form

**Step 1: Insights from the Memory Prompts**

1. **Highest Scoring Item (Score: 25.0)**:
    - **Role**: Sentence Simplifier
    - **Task Description**: Clarify the meaning of the given sentences by simplifying them
    - **Action**: Optimize Readability
    - **Requirements**: Improve Structure
2. **Other Items**:
    - **Role**: Readability Specialist (Score: 21.0)
    - **Role**: Grammar & Clarity Editor (Score: 16.0)
    - **Role**: Technical Content Simplifier (Score: 14.0)

**Step 2: Mutate Prompt 1**

**Mutate Factors**: `<res> role | requirements </res>`

- **Role**: "Sentence Simplifier" is already the highest-scoring role, so we generate a new value: "Clarity Enhancer".
- **Requirements**: "Adapt to Audience" is not the highest-scoring (highest is "Improve Structure"), so we replace it with "Improve Structure".

**Prompt 2**:

```
<prompt>You are a <role>Clarity Enhancer</role>. Given the English Sentence, your task is
to <task_description>Rewrite the provided sentences in a simpler form</task_description>.
To accomplish this, you need to <action>Condense Sentences</action>. Please make sure to
<requirements>Improve Structure</requirements> throughout the process.</prompt>
```

**Step 3: Mutate Prompt 3**

**Mutate Factors**: `<res> role | requirements </res>`

- **Role**: "Linguistic Analyst" is not in the Memory Item List, so we replace it with the highest-scoring role: "Sentence Simplifier".
- **Requirements**: "Preserve Meaning" is not the highest-scoring, so we replace it with "Improve Structure".

**Prompt 4**:

```
<prompt>You are a <role>Sentence Simplifier</role>. Given the English Sentence, your task
is to <task_description>Rewrite the provided sentences in a simpler
form</task_description>. To accomplish this, you need to <action>Condense
Sentences</action>. Please make sure to <requirements>Improve Structure</requirements>
throughout the process.</prompt>
```

**Step 4: Combine Prompt 2 and Prompt 4**

Combine the best-performing markup pairs from Prompt 2 and Prompt 4:

- **Role**: "Clarity Enhancer" (from Prompt 2)
- **Task Description**: "Rewrite the provided sentences in a simpler form" (common in both)
- **Action**: "Condense Sentences" (common in both)
- **Requirements**: "Improve Structure" (common in both)

**Final Prompt**:

```
<prompt>You are a <role>Clarity Enhancer</role>. Given the English Sentence, your task is
to <task_description>Rewrite the provided sentences in a simpler form</task_description>.
To accomplish this, you need to <action>Condense Sentences</action>. Please make sure to
<requirements>Improve Structure</requirements> throughout the process.</prompt>
```

Figure 26: The responses for Sub-solution II - Prompts Memory in continuous form

