# OpenReview forum: "DelvePO: Direction-Guided Self-Evolving Framework for Flexible Prompt Optimization"
_ICLR.cc/2026/Conference — Submitted to ICLR 2026_

### Official Review · Reviewer_akfT · 2025-10-23

**Soundness:** 2
**Presentation:** 3
**Contribution:** 3
**Rating:** 6
**Confidence:** 4

**Summary:**

This paper introduces DelvePO (Direction-Guided Self-Evolving Framework for Flexible Prompt Optimization). DelvePO decomposes prompts into explicit components to systematically probe how different factors affect performance, and augments the process with a working memory that helps the LLM mitigate its own uncertainty, extract key insights, and use those insights to guide the next round of prompt generation. Across diverse tasks and domains, and on both open- and closed-source models (e.g., DeepSeek-R1-Distill-Llama-8B, Qwen2.5-7B-Instruct, GPT-4o-mini), DelvePO consistently outperforms prior state-of-the-art under identical settings, demonstrating improved effectiveness and transferability.

**Strengths:**

1. Decouples prompts into components and uses direction-guided self-evolution, reducing reliance on random rewrites and avoiding local optima while offering clearer interpretability of what matters in a prompt.

2. Introduces a working memory to capture insights across iterations, mitigating LLM uncertainty and making prompt updates more principled and consistent.

3. Demonstrates consistent improvements across diverse tasks and both open/closed models.

**Weaknesses:**

1. Novelty vs structured prompt methods is under-specified. The paper claims a key contribution in decomposing prompts into components, but similar ideas exist (e.g., Task Facet Learning: A Structured Approach To Prompt Optimization). It will be better to add into related works.

2. Missing closer, up-to-date baselines. Need to add some recent and more related prompt-optimization methods as baseline (Task Facet Learning is one example).

3. The transferability claim need stronger evidence. Currently, it is unclear how the transferability is demonstrated in the experiments, which is claimed in the abstract.

**Questions:**

May be better to show some cases with optimized prompt.

---

> ### Author Response · Authors · 2025-11-21
> **Response to Reviewer akfT (Part 1)**
>
> > (a) **W1+W2: Clarification on novelty and baseline comparisons with structured prompt methods**
>
> **Response:** We sincerely thank the reviewer for raising these important points. Since **Weakness 1** and **Weakness 2** are closely related, we provide a unified response below:
>
> - **Review of related work and TFL.**  During our initial literature survey, we were aware of works similar to the Task Facet Learning (TFL) method mentioned by the reviewer. Following your suggestion, we further studied TFL and reproduced its main idea: *decomposing prompts into semantic components, then leveraging **clustering methods** to learn different facets from examples, and finally using LLMs’ internal knowledge to optimize prompts*. In addition, during our early study of prompt-optimization research, we also identified other similar works that use external algorithms to assist LLM-based prompt optimization (e.g., Auto-CoT, EvoPrompt). Due to space constraints, only representative works were included in our related work section. Moreover, TFL primarily reports results on relatively large and powerful base models, while offering only limited evaluation on smaller LLMs. This setting is somewhat misaligned with our target scenario, where we aim to study prompt optimization across LLMs with diverse capability levels. For these reasons, we did not pay much attention on TFL.
>
> - **Comparison with widely recognized baselines.**  To ensure a fair and comprehensive comparison, we selected EvoPrompt and Promptbreeder, two widely recognized and related methods, as our baselines, which also explains why TFL was not included in our original baselines. To better understand TFL, we attempted to reproduce its experiments using smaller-scale models. During this process, we found that due to its design, the practicality of the TFL framework is easily limited to models with relatively small parameter quantities. i.e., TFL may enter an “infinite text generation” loop when calling the LLM, which leads to execution stalling (specifically, lines 20–26 in `api_utils.py`). However, larger models can tolerate this mechanism, highlighting a limitation in TFL’s general applicability. In contrast, DelvePO does not suffer from such issues, demonstrating greater robustness across models of varying sizes.
>
> - **Experimental validation.**  To further address your concern, we ran DelvePO and UniPrompt (the name of algorithm proposed in TFL) on GPT-4o-mini using two commonly used datasets: **Subj** (classical NLP) and **FinPB** (domain-specific). From the results, reported below, we can anticipate that DelvePO continues to achieve competitive performance compared to TFL, reinforcing its generality and effectiveness.
>
> | Methods   | Subj (mean) | FinPB (mean) |
> | --------- | ----------- | ------------ |
> | UniPrompt | 73.08       | 89.42        |
> | DelvePO   | **91.07**   | **98.63**    |
>
> We greatly appreciate your suggestion, which provided a more comprehensive perspective on structured prompt optimization. We will continue to refine our understanding of TFL, perform additional experiments, and update the related work and baseline sections in the revised version. Your insightful comments significantly help us strengthen the clarity and rigor of our work.
>
> > (b) **W3: Clarifying the Evidence for Transferability**
>
> **Response:** We sincerely thank the reviewer for raising this important point. Our use of the term _“transferability”_ follows the definition commonly adopted in automated machine learning literature (e.g., AutoML), where a framework discovered on certain tasks should remain effective when applied to different tasks or settings. In our paper, this concept corresponds to the ability of **DelvePO** to operate across **multiple domains, diverse datasets, and different LLMs** while keeping the **meta-prompts unchanged**, and still consistently improving performance.
>
> More concretely, DelvePO provides a **general, reusable prompt optimization framework** that adapts to:
>
> - different tasks (7 datasets across 4 domains),
>
> - different model preferences (3 LLMs with varying strengths), and
>
> - different user backgrounds, all without redesigning the meta-prompt or manually crafting task-specific strategies. The experimental results reported in the manuscript collectively demonstrate this **strong generalizability across heterogeneous settings**.
>
>
> Moreover, as discussed in our response to **Weakness 03** suggested by Reviewer URXt, we further conducted additional experiments where the target task domain significantly deviates from the initial component set. These supplementary results also indicate that **DelvePO remains effective even under substantial task-domain shifts**, providing further evidence of its robustness across domains.

---

> ### Author Response · Authors · 2025-11-21
> **Response to Reviewer akfT (Part 2)**
>
> (cont.)
>
> In summary, although extensive experiments demonstrate that the DelvePO framework exhibits strong adaptability across multiple tasks, after carefully reconsidering the reviewer’s comment, we agree that the term “transferability” may not be the most precise descriptor. Our findings are better characterized as demonstrating the **generalizability** of DelvePO rather than strict transferability. We will revise the manuscript accordingly and replace the term throughout the paper to avoid ambiguity.
>
> We deeply appreciate the reviewer’s sharp observation, which has helped us refine the presentation of our contributions.
>
> > (c) **Q1: Request to show concrete cases with optimized prompts.**
>
> **Response:**  We thank the reviewer for this helpful suggestion. To better illustrate how DelvePO produces optimized prompts, we provide **two detailed cases** below. The **first case** is available in **[this anonymous document](https://anonymous.4open.science/r/ICLR-DelvePO-Rebuttal/Case%20Study.pdf)**, and for the **second case**, we present and explain it using the examples in our **Case Study** section, where more complete meta-prompts allow readers to trace every step of the optimization process.
>
> ---
>
> **Case 1 (from the anonymous document).**
>
> As shown in the **[anonymous doc](https://anonymous.4open.science/r/ICLR-DelvePO-Rebuttal/Case%20Study.pdf)**, the selected initial prompt contains five components and receives a score of 75. Guided by the current $M_{\text{components}}$, the system identifies ***workflow*** as the component type with the highest potential for performance improvement (i.e., the most promising evolutionary direction). Based on insights summarized from the $M_{\text{prompts}}$, the best-performing value for the workflow component is _“Use sentence structure”_, which is then used to replace the original value.
>
> **A point worth noting is that**: if the current value for the *workflow* component is already the best one stored in $M_{\text{prompts}}$, DelvePO does not simply stop. Instead, the framework encourages the model to **analyze all candidate values in $M_{\text{prompts}}$** from top to bottom, synthesize relevant insights, and generate a **refined, higher-level component value** to replace the existing one. This mechanism enables DelvePO to perform meaningful updates even when the search space appears saturated.
>
> **Case 2 (from the Case Study section).**
>
> A more complete illustration of DelvePO’s pipeline is provided in the Case Study:
>
> - **Mutation of a single prompt (Figure 21 → Figure 22).**  When mutating a single prompt, the Current Prompt consults the $M_{\text{prompts}}$ to identify which factors should be changed. These directions are determined by **Sub-task 1**, and the mutation yields the Final Prompt shown in the upper half of Figure 22.
>
> - **Mutation of two prompts (Figure 25 → Figure 26).**  For two prompts, **Sub-task 2** selects the factors to mutate (Figure 25). DelvePO then applies mutations (Figure 26, Step 2 & Step 3) followed by a crossover operation (Figure 26, Step 4). The resulting optimized prompt is shown in the bottom half of Figure 26.
>
>
> We hope these two cases provide clear, concrete examples of how DelvePO improves prompts through component-memory and prompt-memory guided optimization.
>
> ---
>
> To sum up, we greatly appreciate the positive feedbacks and suggestions. We hope our responses address your concerns and enhance your view of our work!

---

### Official Review · Reviewer_URXt · 2025-10-31

**Soundness:** 2
**Presentation:** 1
**Contribution:** 2
**Rating:** 4
**Confidence:** 3

**Summary:**

The paper proposes DelvePO, a direction-guided and memory-augmented prompt optimization framework. It first decomposes prompts into functional components and then evolves them using two working memories: Component Memory, which records beneficial component edits, and Prompt Memory, which stores high-performing prompt combinations. This design reduces the randomness of LLM-based prompt mutation, prevents the loss of important components, and improves transfer across tasks and models. Experiments on multiple datasets and LLMs show consistent gains over human prompts and prior prompt optimization methods.

**Strengths:**

The paper introduces a clear component-level prompt representation together with two explicit working memories (Component Memory and Prompt Memory), which turns otherwise highly stochastic LLM-based prompt mutation into a more controllable and reusable optimization process.

**Weaknesses:**

1.	The paper proposes a task-agnostic framework, but the initial component pool is manually collected and constructed from a wide range of related literature (line 116). This raises a question about the motivation of the method: does DelvePO truly make no strong task-specific assumptions and generalize to different tasks because of the framework design itself, or is the observed generality mainly due to the fact that a very comprehensive task component pool has been pre-collected and constructed?
2.	The ablation in Table 3 provides only qualitative evidence that the proposed direction signal, instantiated via both the component-level memory and the prompt-level memory, indeed contributes to the final performance: removing either memory degrades the results, and removing both effectively collapses the framework back to a largely stochastic evolution regime. This supports the authors’ claim that “direction-guided” optimization is beneficial. However, earlier in the paper, the authors explicitly state that direction-guided evolution “can reduce the time required for evolutionary operations” (line 87), i.e., that guidance does not merely improve the final score but also makes the optimization more efficient. To substantiate this stronger claim, the paper should report performance vs.–budget curves (e.g., performance as a function of iteration/time, number of LLM calls, or total input tokens) under a fixed budget, and compare them against a purely stochastic/mutation-only variant. Otherwise, the observed gains can still be explained by “doing more (or better-informed) LLM calls” rather than by genuinely improving the sample efficiency of the evolutionary process.
3.	The components used in DelvePO are extracted and constructed from related literature and can be further generated by an LLM. However, there is no experimental evidence showing whether the method remains effective when the target task domain deviates substantially from the initial component set. For domain-specific tasks, you may refer to the test sets used in Table 2 of PROMPTAGENT [1].


[1] Wang, Xinyuan, et al. "Promptagent: Strategic planning with language models enables expert-level prompt optimization." arXiv preprint arXiv:2310.16427 (2023).

**Questions:**

1. The method adopts a multi-stage pipeline whose complexity is relatively high, and the usage cost will be noticeably higher for closed-source LLMs. It is recommended to provide a clearer breakdown of the computational and token cost, and to specify in the method section which stages can be parallelized and which stages can be cached in advance.

2. The authors significantly increase token consumption by writing a large amount of memory back into the prompt, but the experimental section does not systematically report the relationship between “memory context length vs. performance vs. cost.” As a result, it is unclear whether the performance gains are achieved primarily by spending a large number of tokens.

Typos:
1. The double quotation marks around “role” in line 56 are not formatted correctly, and the same issue appears with other quotation marks in the paper.

---

> ### Author Response · Authors · 2025-11-21
> **Response to Reviewer URXt (Part 1)**
>
> > (a) **W1: Clarifying the task-agnostic nature of DelvePO vs. the role of the initial component pool**
>
> **Response:** We sincerely thank the reviewer for raising this important question regarding whether DelvePO’s task-agnostic generality arises from the framework design itself or mainly from the comprehensively collected initial component pool.
>
> **Firstly**, we would like to clarify that **DelvePO makes no task-specific assumptions**, and its generalization ability is rooted in the framework design. This is reflected in **two aspects**:
>
> 1. **Consistent framework across tasks:** The same DelvePO framework is applied to all datasets and tasks. Users **do not need to** modify any meta-prompts based on the task.
>
> 2. **Identical initial component set across tasks:** Although we provide a component pool, the same initial components are used for all tasks reported in our experiments. The only task-dependent difference is in the candidate values generated from these components, which DelvePO automatically produces based on the task description—**no manual modification is required**. (We provide a detailed explanation of the component set in our response to **Question 03** from Reviewer ga5y.)
>
>
> **Second**, from experimental results across **three LLMs** and **seven datasets** in **four domains**, it is evident that the framework, rather than the component pool, drives the observed generality. The component pool serves merely as a **candidate repository** to provide convenient initial values for users, **rather than** introducing task-specific knowledge.
>
> Finally, we would like to emphasize the **core motivation** of DelvePO: As the variety of LLMs and application scenarios continues to grow, DelvePO aims to provide a **lightweight, task-agnostic prompt optimization framework** that enables users to optimize prompts effectively for their tasks, regardless of the specific LLM used.
>
> We deeply appreciate the reviewer’s careful observation, which helps clarify the key contribution of our work, and we hope this response satisfactorily addresses your concern.
>
> > (b) **W2. Request for quantitative evidence of direction-guided efficiency (performance–budget analysis)**
>
> **Response**: We are glad to see that reviewer URXt **recognizes the effectiveness** of the DelvePO framework. To provide a more comprehensive and detailed validation of DelvePO’s performance in prompt optimization, we have integrated this part of the discussion with the response to **Question 2** raised by **reviewer y3qT**. The reviewer is kindly directed to the corresponding responses. We sincerely appreciate your insightful perspectives, which helped us refine both the analysis and presentation.
>
> > (c) **W3: Concern about DelvePO’s generalizability when task domains deviate from the initial component set**
>
> **Response:** Thank you for raising this important point. We understand your concern regarding the generalizability of our framework when facing domain-specific tasks. We address this issue from two perspectives:
>
> - **Transferability and flexibility of components.**  As detailed discussion in our response to **Weakness 01** you mentioned above, the components used in DelvePO indeed exhibit strong transferability. In summary, all datasets used in our experiments rely on the same predefined component set and meta-prompts, without requiring any task-specific modification from the user. The consistent performance across three models of different capability levels demonstrates that DelvePO provides strong general applicability. More importantly, one of the key contributions of our work is offering a **general, forward-looking framework** in which component definitions are flexible and easily adjustable. Users may **directly adopt** our predefined components **when no specific guidance is provided**, or—if domain-specific factors are required—**can customize** components based on our component formulation. DelvePO can then leverage these adapted components to optimize prompts for the target tasks. From all reported experiments, the predefined component set proves to be effective.
>
> - **Additional experiments on the domain-specific tasks you mentioned.**  In the main experiments, our datasets span four domains and seven datasets of varying difficulty (Finance, News, QA, Multi-domain, and classical NLP). Compared with baselines, **the domains covered by our datasets are already quite diverse**. To further address your concern, we additionally evaluate DelvePO on **two randomly selected datasets** from those referenced in Table 2 of PromptAgent. The results for GPT-3.5-Turbo are below:
>
> | Baselines   | Biosses             | Med QA              |
> | ----------- | ------------------- | ------------------- |
> | Ape         | $70.00$             | $47.00$             |
> | EvoPrompt   | $70.20$             | $\underline{65.00}$ |
> | Promptagent | $\underline{75.00}$ | $57.00$             |
> | DelvePO     | $\textbf{76.40}$    | $\textbf{86.30}$    |

---

> ### Author Response · Authors · 2025-11-21
> **Response to Reviewer URXt (Part 2)**
>
> (cont.)
>
> From the additional experiments, we can anticipate that DelvePO continues to achieve **competitive performance** compared to PromptAgent, even on domain-specific tasks where the target domain substantially deviates from the initial component set. This observation is consistent with our earlier claims and provides further evidence of the robustness and effectiveness of the proposed framework.
>
> Due to time constraints, we will continue running more extensive experiments and will include the updated results in the revised version. We appreciate your valuable suggestion, which greatly helps strengthen the empirical support for DelvePO’s generalizability.
>
> > (d) **Q1: On the computational complexity, token cost, and opportunities for Parallelization/Caching**
>
> **Response:** We are so grateful to reviewer URXt for the thoughtful and constructive comments regarding the computational and token cost of DelvePO. We fully agree that understanding the cost breakdown and identifying opportunities for optimization are important for assessing the practicality of our approach. We address the reviewer’s concerns from three complementary perspectives:
>
> **(1) Cost vs. performance trade-off.**
>
> As shown in **Table 6** in the original manuscript, DelvePO incurs a slightly higher cost compared with baselines. However, when jointly considering **Table 2** and **Table 6**—which report performance and cost on the _exact same datasets_ (split for space reasons)—DelvePO achieves an **average improvement of nearly 5 points**, comparable to the gains we observe on DeepSeek-R1-Distill-Llama-8B (Table 1), indicating stable performance benefits across LLMs and domains. From a cost–effectiveness perspective, the increased expenditure remains **tolerable** given the consistent and substantial performance improvements.
>
> **(2) Cost and performance on closed-source LLMs.**
>
> We acknowledge the reviewer’s concern that the cost may be more noticeable for closed-source models. Empirically, however, the results in our closed-source LLM experiments (detailed in the response to **Question 2** for reviewer **y3qT** ) show that DelvePO continues to outperform all baselines under the same conditions, suggesting that the framework remains effective.
>
> **(3) Additional experiments motivated by the reviewer’s comments.**
> Inspired by the reviewer’s insightful suggestions (and also reviewer **y3qT’s Q2**), we conducted **new experiments** analyzing:
>
> - performance as a function of token cost, and
>
> - convergence behavior under fixed token budgets.
>
>
> Across these experiments, DelvePO consistently reaches **convergence in fewer epochs** and achieves **higher final performance** than baselines under the same budget constraints. These findings not only demonstrate enhanced cost–performance efficiency but also highlight the **effectiveness of the memory mechanism** in reducing redundant exploration.
>
> We hope these results collectively address the reviewer’s concerns regarding computational cost.
>
> ---
>
> **Parallelization Opportunities**
>
> We appreciate the reviewer for pointing out the potential to reduce runtime via parallel execution. After analyzing the workflow of DelvePO, we found that **task evolution** and **solution evolution** within each iteration are independent once the candidate individuals are selected. These steps have **no mutual dependencies** and can therefore be parallelized, reducing wall-clock time without altering the algorithmic logic. We will make this clearer revised method section.
>
> ---
>
> **Caching Opportunities**
>
> The reviewer’s suggestion regarding caching is extremely insightful. Building upon our memory mechanism, we find two natural opportunities for caching:
>
> - **Caching optimized prompts:**  Before applying evolution, our framework can query whether an identical or similar prompt has already been evaluated. This avoids duplicated evolutionary operations.
>
> - **Caching evaluation results:**  Since evaluation can be relatively costly for closed-source LLMs, caching previously optimized prompts allows us to avoid re-evaluating them, directly reducing **repeated token consumption**.
>
>
> Both caching strategies would meaningfully reduce redundant computation and further improve efficiency. We thank the reviewer a lot for highlighting this direction, which provide a strong foundation for future research and practical applications in PO area.
>
> ---
>
> We sincerely appreciate the reviewer’s valuable feedback. These valuable suggestions not only help clarify our current cost structure but also open up promising avenues to make DelvePO more efficient and practical. We are committed to further exploring these solutions and sharing our findings among the community. Thank you again for your insightful feedback.

---

> ### Author Response · Authors · 2025-11-21
> **Response to Reviewer URXt (Part 3)**
>
> (cont.)
>
> > (e) **Q2: Systematically report on the relationship between performance and cost**
>
> **Response**: We appreciate the reviewer’s insightful question regarding the relationship between performance and cost. In fact, the performance gains of **DelvePO** do **not** come from accumulating large memory tokens. The token-related cost of DelvePO mainly comes from three components: **meta-prompts**, **memory context** (component memory & prompts memory), and basic **input/output**. The role of memory in DelvePO is to _guide task evolution toward promising component types_ and to _provide insights for solution evolution_ to perform evolutionary operations. Importantly, these memories are **not directly fed to the LLM as prompts**. Specifically, throughout the entire process, the memory itself does not serve as a prompt to directly guide the LLM. Instead, it is used only during the early interaction phases of task evolution and solution evolution to help the LLM extract useful insights. These insights then act as a directional signal, enabling the LLM to identify promising evolutionary directions and operations based on the current individuals. Moreover, the memory size is strictly bounded—in our implementation, it is limited to the same size as the population—ensuring that the memory does not accumulate indefinitely, which means that DelvePO does **not** rely on token inflation to improve performance. Instead, its improvements stem from the proposed memory mechanisms combined with iterative evolution of task- and solution-.
>
> To address potential overlapping concerns as clearly as possible, the relationship between performance and cost has been **theoretically analyzed** and **empirically validated** in previous comments. We kindly refer the reviewer to our detailed response to **Question 02** raised by **y3qT**, where all related discussions are fully elaborated.
>
> We sincerely appreciate the reviewer for raising such a thoughtful comment on **performance–cost trade-offs**, which helped us clarify this important aspect of our method.
>
> > (f) **Q3: On the Incorrect Formatting of Quotation Marks**
>
> **Response:** We are grateful to the reviewer for carefully pointing out the formatting issue with the double quotation marks in line 56, as well as similar issues elsewhere in the manuscript. We apologize for the oversight. We have now **thoroughly reviewed the entire paper** and will **correct all quotation mark usages** in the revision.
>
> Specifically, we will replace all improperly formatted quotation marks with the correct LaTeX forms:
>
> - left double quotes: $\texttt{``}$
>
> - right double quotes: $\texttt{''}$
>
>
> We appreciate the reviewer’s attention to detail, which helps improve the clarity and readability of our manuscript.

---

### Official Review · Reviewer_y3qT · 2025-11-01

**Soundness:** 2
**Presentation:** 3
**Contribution:** 2
**Rating:** 4
**Confidence:** 4

**Summary:**

Previous works about prompt optimization focuses on limited specific factors, making local optimum inevitable. This work proposes DelvePO with memory to recognize deficiencies and thus guide new generation of prompt. DelvePO consistently outperforms previous SOTA methods on classical NLP tasks, QA, etc. across several models, showing the effectiveness of DelvePO. Specifically, DelvePO decouples the prompt into several factors, task-evolution, solution-evolution, memory-evolution, to guide the evolutionary process. The integration of multiple modules could also improve the interpretability of the optimization process, making it easier to interact with the  system.

**Strengths:**

* DelvePO achieves better performance compared with previous baselines and ablation study shows the effectiveness of each component in the method.
* This paper is well-written and the motivation is clear, meaningful.

**Weaknesses:**

* Datasets and tasks selected are classical, relatively easy tasks for LLMs and these are not difficult for current strong LLMs anymore. I'm curious about the performance of DelvePO on more challenging and difficult tasks in LLM-era, like GSM8k, BBH, more reasoning tasks and so on.
* This paper introduces memory and in essence, memory appears as concluding insights from last-generation prompts, which is a little far-fetched. OPRO[1] gives previous good-performing prompts and worse prompts to guide generation, APO[2] gives bad cases to guide optimization, all of which can be explained as "memory". I disagree with the statement "DelvePO is the first one to introduce memory", instead, it seems that DelvePO first explains such optimization as "memory".
* Though the motivation is meaningful, I don't think DelvePO solves the problem pointed out, i.e. local optimum. Jumping out of local optimum has not been proved quantitively, I'm not convinced of the claim.
## references
[1] Yang, Chengrun, et al. "Large language models as optimizers." The Twelfth International Conference on Learning Representations. 2023.
[2] Pryzant, Reid, et al. "Automatic Prompt Optimization with" Gradient Descent" and Beam Search." The 2023 Conference on Empirical Methods in Natural Language Processing.

**Questions:**

* Could the authors provide ad detailed example of $M_{components}$
* In table 6, the cost of DelvePO is relatively high. The experimental proformances compared with baselines under same costs should be investigated further.
* See the weakness part.

---

> ### Author Response · Authors · 2025-11-21
> **Response to Reviewer y3qT (Part 1)**
>
> > (a) **W1: Concern about the lack of evaluation on more challenging LLM-era reasoning tasks**
>
> **Response:** Thank you for raising this point. We address the concern about dataset selection and the difficulty level of the tasks below:
>
> - **Dataset selection criteria.**  Our choice of datasets follows two principles: _fair comparison_ and _community adoption_. To ensure **fairness in comparison with prior PO studies**, we use datasets that are widely adopted in existing PO literature. As detailed in the Appendix B, these datasets vary in difficulty and cover multiple domains, and they have been **commonly recognized benchmarks** in PO related research.
>
> - **Capability of the models used.**  We agree with your observation that “classical, relatively easy tasks” may no longer be challenging for strong LLMs. In our experiments, however, two of the three models we evaluate—**DeepSeek-R1-Distill-Llama-8B** and **Qwen2.5-7B-Instruct**—are models with **relatively small parameter scales and moderate capability**, rather than **the strong contemporary LLMs**. The results on these models show that DelvePO consistently outperforms existing baselines across diverse domains.
>
> - **Additional experiments on more challenging tasks.**  To further address your concern, we additionally evaluate DelvePO on GSM8K and two randomly selected BBH tasks (**Casual Judgement** and **Ruin Names**), in which Casual Judgement is a commonly used reasoning dataset and is representative of LLM-era challenging tasks. The results are shown below:
>
> |           | Causal Judgement | Ruin Names | GSM8K    |
> | --------- | ---------------- | ---------- | -------- |
> | APE       | 57.86            | 57.54      | 81.6     |
> | EvoPrompt | 67.14            | 53.77      | 82.6     |
> | DelvePO   | **69.29**        | **57.79**  | **83.2** |
>
> These results indicate that DelvePO remains effective on _more challenging and difficult tasks_, demonstrating robust improvements over baselines. **Taken together with the above explanations and the additional experiments**, we can anticipate that DelvePO achieves effective prompt optimization across multiple domains and under LLMs of different capability levels, demonstrating its strong generalizability and effectiveness. We appreciate your question and hope our response resolves your concerns.
>
> > (b) **W2: Clarifying the use and definition of "Memory" in DelvePO**
>
> **Response:** We thank the reviewer for the insightful comment regarding our **use** of “memory” in DelvePO and its relation to prior works such as OPRO and APO.
>
> In current research, particularly in the context of agent-based methods [1], there appears to be **no widely agreed-upon definition of memory**. In this work, we draw inspiration from **human learning and reasoning paradigms**, defining memory as a **knowledge repository that is accumulated over the learning process, can be repeatedly used, and continuously updated**.
>
> We note that OPRO, APO, and similar works can also be interpreted as leveraging memory: they use previous high- or low-performing prompts to guide generation or optimization. However, as noted by Shum et al. [2], such case-based approaches often suffer from limited generalization due to the scarcity of suitable examples. In contrast, DelvePO formalizes this concept as an **explicit and operational memory module**, which guides both task evolution and solution evolution. We further validate the effectiveness of this module through ablation studies, demonstrating that combining components with the memory mechanism enables faster and more robust prompt optimization across multiple datasets and LLMs.
>
> We recognize that the idea of memory is not entirely new, but DelvePO’s contribution lies in **explicitly structuring and reinforcing it as a module within prompt optimization**, which, to our knowledge, has not been done before. To improve clarity, we will include a formal definition of memory in the **Preliminary** section of the revised manuscript and will clarify its usage consistently throughout the text.
>
> We greatly appreciate the reviewer’s careful observation, which helps us refine the presentation and enhance the rigor of our work.
>
> > [1] Luo, Junyu, et al. "Large language model agent: A survey on methodology, applications and challenges." _arXiv preprint arXiv:2503.21460_ (2025).
> >
> > [2] Shum, KaShun, Shizhe Diao, and Tong Zhang. "Automatic prompt augmentation and selection with chain-of-thought from labeled data." _arXiv preprint arXiv:2302.12822_ (2023).
>
> > (c) **W3: Clarifying DelvePO’s perspective on the term “local optimum” in PO area**
>
> **Response:** We thank the reviewer for raising this insightful question regarding the issue of local optima in prompt optimization.

---

> ### Author Response · Authors · 2025-11-21
> **Response to Reviewer y3qT (Part 2)**
>
> (cont.)
> > We'd like to kindly remind reviewers to check some **new figures** we provided through **[anonymous links](https://anonymous.4open.science/r/ICLR-DelvePO-Rebuttal/)** embedded in the text (cannot directly embed them into the responses).
>
> In our manuscript, the term _“local optimum”_ in Prompt Optimization refers to a common **phenomenon**: inappropriate optimization strategies can lead to the **loss of important prompt components**, which cannot be recovered, ultimately resulting in the prompt performance remaining at a relatively low level without further improvement. We use this terminology in PO by analogy to the concept of local optima in machine learning. For example, as shown in **Figure 1 (presented in EvoPrompt)**, once a critical component is discarded during evolution, it cannot be recovered, resulting in stagnated performance. Our experiments find that these lost components often **have a significant impact on the final prompt quality**, whereas DelvePO consistently outperforms EvoPrompt by guiding the evolution process and retaining key components.
>
> In summary, conventional evolutionary algorithms alone typically fail to address this phenomenon mentioned above. In contrast, DelvePO provides **directional guidance** for the evolutionary process, helping optimize prompts more efficiently. Experimental results across multiple datasets and LLMs further demonstrate that DelvePO effectively mitigates the adverse effects of local optima in practice.
>
> We also note that **quantitative proof of “jumping out of local optimum” is not yet feasible** in the prompt optimization domain. We apologize for any confusion caused by our previous wording and will revise the manuscript to clarify these statements. We greatly appreciate the reviewer’s careful observation, which helps us refine the presentation of this important aspect of our work.
>
> > (d) **Q1: Request for a detailed example of $M_{\text{components}}$**
>
> **Response:** We thank the reviewer for the question. Due to space limitations in the original manuscript, the detailed examples of $M_{\text{components}}$​ were placed in the Appendix (**Figure 15** and **Figure 17**). To directly address your request, we provide a complete example—covering $M_{\text{components}}$​, $M_{\text{prompts}}$​, the **selected component types** for mutation (i.e., promising mutation directions), and the **prompts** before and after evolution—via **[an anonymous document](https://anonymous.4open.science/r/ICLR-DelvePO-Rebuttal/Case%20Study.pdf)**.
>
> We also noticed that this question is included in **Question 01** raised by reviewer akfT. A detailed explanation has been provided in our response to akfT-Q1 (see **the first case**). And for your convenience, we offer a brief summary of the role of $M_{\text{components}}$ below.
>
> **Summary of the role of $M_{\text{components}}$​**: $M_{\text{components}}$​ is primarily used in **Sub-tasks**. Its goal is to extract insights from the prompts before and after evolutionary, helping identify which component types should be mutated—i.e., those that are most **promising for improving the prompt performance**.
>
> More specifically:
>
> - As shown in Figure 15 and Figure 17 (two representative snapshots), $M_{\text{components}}$ is stored as a **List** that is continuously updated throughout the iterations. Each element is a **pair**, where the first component outperforms the second one. Thus, the pairs appear in descending performance order.
>
> - During each iteration, the LLM interacts with $M_{\text{components}}$​ to determine which component types are most likely to yield beneficial mutations. These selected component types then guide the corresponding **Sub-solutions** to execute evolutionary operations.
>
> We hope the provided example and explanation clarify the functionality of $M_{\text{components}}$. Following suggestions from multiple reviewers (ga5y-Q3, akfT-Q1), we will reorganize and refine the presentation of **Case Study** section in the revised manuscript. We sincerely appreciate your valuable feedback, which greatly helps enhance the clarity and completeness of our work.
>
> > (e) **Q2: Comprehensive and detailed analysis about Performance–Cost concern of DelvePO**
>
> **Response**: We sincerely thank reviewers **y3qT**, **URXt**, and **ga5y** for raising insightful and valuable comments regarding the performance–cost relationship. We also apologize for the **typos** in the **caption of Table 6**, where _one epoch_ should be corrected to _three runs_. Although this mistake does not affect the validity of comparison, it unintentionally overstated our reported cost (since _3 runs = 30 epochs_). We identified this issue while re-running the performance–budget experiments and will correct it in the revision. This may have contributed to the impression that DelvePO incurs unusually high cost. Below, we present a consolidated and thorough response.

---

> ### Author Response · Authors · 2025-11-21
> **Response to Reviewer y3qT (Part 3)**
>
> (cont.)
>
>
> - **Why DelvePO appears more costly**: After revisiting related comments (ga5y-W4, URXt-W2 and URXt-Q2), we carefully examined why DelvePO’s reported cost occasionally exceeds that of baselines, and found that the original experimental setup relied solely on a fixed-epoch stopping criterion. In several tasks, **DelvePO had already discovered its optimal prompts early**, but the optimization procedure still continued until the epoch limit was reached. This unnecessary “idle running” accumulated extra tokens and inflated the apparent cost.
> - **Additional evidence from performance–budget experiments**: Following the reviewers' advice, we conducted **new performance-vs.-budget experiments** on two representative datasets—**Subj** (**[budget $0.4](https://anonymous.4open.science/r/ICLR-DelvePO-Rebuttal/Subj,%20cost%20fixed%20to%20$0.4.png)**) and **SAMSum** (**[budget $0.6](https://anonymous.4open.science/r/ICLR-DelvePO-Rebuttal/SAMSum,%20cost%20fixed%20to%20$0.6.png)**). We compared DelvePO with baselines under **two stopping conditions**:  (1) reaching the epoch limit, and (2) exhausting the cost budget. For **Subj** (budget $0.4):  DelvePO surpasses all baselines at **epoch 4**, reaches its peak around **epoch 6**, and remains stable thereafter.  This demonstrates that: 1) **DelvePO converges rapidly** under the guidance of its memory mechanism—consistent with the time-efficiency trends reported in Figure 3 at original manuscript; 2) The additional cost arises mainly from **continued epochs after convergence**, matching our earlier analysis.  A similar pattern is consistently observed on **SAMSum**  (budget $0.6).
> - **Experiments under different budgets**: We further evaluated performance under **multiple cost budgets**. The results on **[Subj](https://anonymous.4open.science/r/ICLR-DelvePO-Rebuttal/Subj,%20cost%20vs%20scores.png)** and **[SAMSum](https://anonymous.4open.science/r/ICLR-DelvePO-Rebuttal/SAMSum,%20cost%20vs%20scores.png)** reveal that: 1) DelvePO achieves **higher final performance** than baselines even when consuming more tokens. 2) Baselines such as APE and EvoPrompt tend to **prematurely plateau** at suboptimal solutions, and EvoPrompt frequently shows **noticeable instability**—a behavior also observed on SAMSum and consistent with our response to **Weakness 3**. 3) Besides, on SAMSum, **PromptBreeder fails to improve at all** under a budget of $0.2 (score remains 0), supporting our observation reported in ga5y-Q3.
>
> **Summary**: Across all analyses, our findings clearly indicate that:
> - Under the **same budget**, DelvePO converges **faster** and reaches **higher peak performance** than baselines.
>
> - Under **various budgets**, DelvePO consistently provides **stable and superior prompt optimization**.
>
> - The **typos** in caption of Table 6 unintentionally exaggerated our cost, but after correction, DelvePO remains **highly competitive** in performance, cost-efficiency, and time consumption.
>
> - Given its **flexible design** and the rapid progress of LLMs, we believe DelvePO is well-positioned to deliver increasingly efficient, scalable, and high-quality prompt optimization in real-world applications.
>
> We deeply appreciate the reviewers’ thoughtful perspectives on this topic. We hope our clarifications fully address all concerns, and we will incorporate these insights carefully into the revised manuscript. We are grateful for the opportunity to further strengthen our work.

---

> > ### Comment · Reviewer_y3qT · 2025-11-27
> > **Response to Authors' rebuttal**
> >
> > I thank the authors for their extensive results and clarifying explanations. Now I have a more clear understanding about memory. Basically, DelvePO decouples the prompt into several aspects including workflow, output format, etc. as memory called. I do not have other questions and I'd like to keep my score.

---

### Official Review · Reviewer_ga5y · 2025-11-01

**Soundness:** 2
**Presentation:** 2
**Contribution:** 2
**Rating:** 4
**Confidence:** 3

**Summary:**

This paper proposes DelvePO, a framework for automatic prompt optimization that decouples prompts into modular components (analogous to genetic loci and alleles) and uses a working memory mechanism to guide evolutionary operations. The framework consists of three main modules: Task-Evolution (determining which components to evolve), Solution-Evolution (performing mutation/crossover operations), and Memory-Evolution (updating component and prompt memories). Experiments are conducted on 11 datasets across 3 LLMs (DeepSeek-R1-Distill-Llama-8B, Qwen2.5-7B-Instruct, GPT-4o-mini), showing improvements over baselines including APE, PromptBreeder, and EvoPrompt.

**Strengths:**

- Decomposing prompts into interpretable components is valuable for understanding what makes prompts effective
- Testing across multiple LLMs and domains demonstrates effort to validate generalizability
- Detailed appendices with all prompts used enhance transparency
- The working memory design that stores both component-level and prompt-level information is sensible

**Weaknesses:**

- The core contributions are incremental improvements over existing evolutionary prompt optimization methods
- Lack of significance testing and inconsistent use of random seeds weakens confidence in reported improvements
- The framework requires extensive prompt engineering (Sub-tasks I-II, Sub-solutions I-II, multiple scenarios) that may limit adoption
- Practical Limitations:

1. Higher computational costs than baselines
2. Requires predefined component types that may not transfer across domains
3. The "case study" reveals manual steps that contradict automation claims

**Questions:**

- Can you provide statistical significance tests (e.g., paired t-tests) comparing DelvePO against baselines across random seeds?
- Table 4 shows concerning instability with different numbers of component values. How do you recommend practitioners set this hyperparameter?
- The case study (Appendix I) involves manual interaction with DeepSeek Chat. How does this square with claims of a fully automated framework?
- Can you provide ablations showing the contribution of individual components to overall performance? Which components are most important?
- How does performance scale with the number of component types? What happens if users define 10+ components?
- The discrete vs. continuous prompt memory distinction is unclear. Can you provide empirical comparison of these two approaches?
- Why were different subsets of datasets used for different LLMs? This makes it difficult to draw conclusions about model-specific behaviors.
- How sensitive is the method to the quality of initial component value generation (Figure 4)?
- Can you provide analysis of failure modes? When does DelvePO underperform simpler baselines?
- The framework requires many carefully crafted meta-prompts (Figures 8-14). How much prompt engineering effort went into developing these, and how transferable are they?

---

> ### Author Response · Authors · 2025-11-21
> **Response to Reviewer ga5y (Part 1)**
>
> > (a) **W1: Clarification about core contributions**
>
> **Response**: We appreciate the reviewer for pointing this out. While our work is indeed built upon the general paradigm of evolutionary algorithms (EAs), it is **not** merely an incremental refinement of prior EA-based prompt optimization methods. As noted in the paper, existing approaches such as **PromptBreeder** and **EvoPrompt** both adopt evolutionary frameworks, yet they focus on different aspects: EvoPrompt directly applies a standard EA pipeline to prompts rewording, whereas PromptBreeder works on the improvement of specific evolutionary operators while largely preserving the overall structure.
>
> In contrast, **DelvePO introduces two key innovations that go beyond simple algorithmic adjustments**:
>
> 1. **Componentization of prompts:**
>     We propose the notion of _prompt components_, which allows users from different domains to explicitly define and customize the domain-critical factors for optimization. This design turns the EA pipeline into a flexible and extensible framework that can be tailored by domain experts, rather than a fixed mutation-only process.
>
> 2. **A memory-driven optimization mechanism:**
>     Building on components, we introduce a memory mechanism that accelerates convergence and more effectively identifies high-quality prompts than existing EA-based approaches. This mechanism helps our method adapt to different LLMs, balance exploration and exploitation in the evolving prompt population, and leverage the strengths of each model more fully.
>
>
> Thus, unlike prior methods that rely primarily on mutation-based evolutionary search, DelvePO combines componentization with memory in a way that **systematically enhances the optimization capacity of evolutionary algorithms**, particularly under scenarios shaped by rapid LLM advancement. Our experimental results consistently show that DelvePO outperforms SOTA EA-based prompt optimization baselines across different models and tasks.
>
> We believe that as future LLMs become increasingly heterogeneous and domain-specialized, DelvePO’s design will enable it to evolve correspondingly, further strengthening its adaptability.
>
> > (b) **W2 & Q1: Concerns about stability of experimental performance**
>
> **Response**: We thank the reviewer for raising this concern. We understand that your question stems from uncertainty about the stability of our experimental results, and we address it from two perspectives below:
>
> - **Statistical significance test.** Regarding significance testing, this issue closely aligns with Question 1 you mentioned, and we would like to offer a unified and detailed explanation here:
> 	- Before designing our experiments, we conducted a broad survey of prior PO-related work with the specific question of how stability is typically demonstrated. We found that nearly all existing studies acknowledge a practical challenge: prompt optimization is highly resource-intensive, and running a large number of repeated trials incurs substantial computational cost (both in time and budget). Consequently, the standard and widely adopted practice is to use **a small number of random seeds (typically 3~5)** and report the **mean and standard deviation** of the results (e.g., Figure 7 in APE, Table 1 in Section 4.1 of EvoPrompt). To mitigate randomness, our experimental design follows the same convention.
> 	- We would also like to note that, prior to your comment, we attempted to apply statistical significance testing (e.g., paired *t*-tests) to further support the stability of our results. However, because the number of seeds cannot be large in PO settings, the resulting sample size is very limited (giving a degree of freedom of only **df = 2**). This leads to reduced statistical power in the *t*-test, making the *p*-value more likely to exceed 0.05 even when performance improvements are consistent. In such cases, the test could misleadingly fail to reflect the effectiveness of the method.
> 	- For these reasons, we ultimately followed the standard practice of reporting **mean ± standard deviation**, which remains the most reliable and commonly accepted way to account for randomness under the computational constraints of PO research.
>
> - **Explanation about random seeds**. On the matter of random seeds, our settings are in fact consistent. Random seeds are used primarily to assess robustness. In line with common practice in the prompt optimization literature, **manually designed baselines are typically run once**, and we follow this convention. For **automated methods** (unless otherwise noted), including DelvePO, we adhere to the baseline protocols and run experiments with **three random seeds (5, 10, 15)**. The aggregated results are reported in **Tables 1, 2, and 5**. From both the reported results in the tables, we can observe that the performance of our method is indeed robust.

---

> ### Author Response · Authors · 2025-11-21
> **Response to Reviewer ga5y (Part 2)**
>
> (cont.)
>
> Upon re-examining the manuscript, we realized that our description of the experimental setup was unintentionally condensed and placed in the _Reproducibility Statement_, which may have led to this misunderstanding. Your comment helped us recognize that these details warrant clearer emphasis. We will expand and clarify this section in the revision to ensure that the experimental protocol and seed usage are fully transparent.
>
> We sincerely appreciate your constructive feedback and hope our response addresses your concern.
>
> > (c) **W3: Concern about the extent of prompt engineering required**
>
> **Response**: Thank you for raising this question. We fully understand the reviewer’s concern that extensive prompt engineering could potentially limit practical adoption. A more detailed discussion is provided in **ga5y – Q8**; here, we summarize the key points most relevant to your comment:
>
> - Although the framework conceptually introduces several meta-prompts (for Sub-task I–II and Sub-solution I–II), **the main experiments employ a single, fixed, and unified set of meta-prompts**. These meta-prompts only need to be designed **once**, are **integrated into the framework**, and **are never exposed to end users**. In other words, users are **not** required to manually craft or adjust these meta-prompts during actual use.
>
> - Moreover, our experimental results indicate that this unified set of meta-prompts **generalizes robustly across different LLMs and tasks**. Across all evaluation settings, DelvePO consistently outperforms baselines **without any modification** to the meta-prompts. This suggests that the framework maintains strong transferability and **does not impose additional overhead on users or impede practical adoption**.
>
> > (d) **W4: Practical Limitations**
>
> **Response**: Thank you for these precise and thoughtful comments. Below we address each of the three practical concerns in detail.
>
> **1. Computation vs. Performance Balance**
>
> We acknowledge that DelvePO involves slightly greater computational overhead compared with some baselines. However, this overhead to some extent balanced by its **substantial gains in both performance and optimization speed**, resulting in a more efficient overall process than the baselines.
>
> To further address this concern, and following the insightful suggestions from reviewers y3qT (Question 2) and URXt (Weakness 2), we conducted an additional experiment evaluating performance **under equal computation budgets**. As detailed in our response to reviewer y3qT (Question 2), these results show that **DelvePO surpasses the baselines in fewer epochs under constrained computation budgets**, indicating that its optimization efficiency is understated by the epoch-based comparisons in the main paper. Because our stopping criteria followed the baselines (fixed epochs), these equal-budget results were not included in the initial submission.
>
> The new results confirm that DelvePO is superior in **performance, time consumption, and equal-budget efficiency**. We sincerely appreciate your and the other reviewers’ feedback, which helped us recognize the need to emphasize this point. We will update the revised version accordingly.
>
> ---
>
> **2. Transferability of predefined component types**
>
> We noticed that this issue also appears in **Weakness 3** and **Question 3**, and we provide detailed explanations there. Here we summarize the key points relevant to your concern:
>
> - DelvePO’s **meta-prompts and predefined component sets are designed once and reused across all tasks**.
>
> - In all main experiments—including **4 domains and 7 datasets**—we use the _same_ set of components and meta-prompts, **without any task-specific customization**.
>
> - Despite this, DelvePO consistently outperforms baselines, demonstrating that the predefined components **transfer effectively across domains**.
>
>
> This strong empirical generalization indicates that the component types we propose are not domain-specific but rather broadly applicable.
>
> ---
>
> **3. Clarifying on the claim about manual interaction vs. fully automated framework**
>
> We thank the reviewer for pointing out this important detail regarding **Appendix $\textbf{I}$**. We agree that the current wording may lead to confusion regarding our claim of a **fully automated framework**. In summary, the reported case study is fully consistent with the automated optimization framework. Upon reviewing the manuscript, we realized that our explanation for using a web-based model was brief, and some descriptions lacked sufficient background, which may have caused confusion. We apologize for this and provide a detailed clarification below:

---

> ### Author Response · Authors · 2025-11-21
> **Response to Reviewer ga5y (Part 3)**
>
> (cont.)
>
> The primary purpose of the case study is to allow readers to **quickly validate the effectiveness of the framework**. Unlike typical algorithmic frameworks, prompt optimization (PO) work has a unique advantage: besides experimental reproduction, one can **intuitively verify framework effectiveness using web-based LLMs**. This motivated our design of the case study.
>
> In the main text, we evaluated DelvePO on **three LLMs**, and the results show that our method is **model-agnostic**. To help readers quickly verify the framework, and considering reviewers’ and readers’ busy schedules, the case study demonstrates a **snapshot of DelvePO’s optimization process** on an easily accessible web-based LLM. The focus is on **intuitively illustrating the contribution of each mechanism** (task-, solution- and memory-evolution) during the optimization. Importantly, for any given target task, the algorithm’s **inputs and outputs are fully generated automatically** by DelvePO.
>
> More precisely, the case study serves as a **rapid prototype**, with each step representing a frame of DelvePO’s full operation. For further clarification, we would kindly redirect the reviewer to our response to **akfT (Question 1)**, where we present a detailed explanation about **complete optimization process along with the corresponding results**.
>
> Concerning the matter you raised, we will subsequently refine the structure of the case study section and offer a detailed explanatory note below the relevant results in the **Appendix $\textbf{I}$**. We appreciate your constructive comments.
>
> > (e) **Q2: Guidance on setting the number of component values**
>
> **Response**: Thank you for the question. Before addressing it directly, we would like to clarify several key concepts from the paper **to ensure precise terminology in our explanation**:
>
> - **Component type:** e.g., _role_, _task description_, … We form a component pool containing various types of components, which can be found in **Table 7 (Appendix E)** of the manuscript.
>
> - **Component value:** the actual value that a component type can take. For example, for the type _role_, one possible value is _math teacher_ when solving math-related tasks.
>
> - **Number of component types:** set to **5** in our paper—_role_, _task description_, _output format_, _workflows_, and _examples_ (also shown in Table 7).
>
> - **Number of component values:** the number of candidate values that a particular component type can draw from; this is set to **10** in our work.
>
>
> We fully understand your concern regarding the instability that appears when varying the **number of component values**. In fact, this is precisely what Table 4 is designed to investigate. Our choice of **10** as the default was informed by the observations in Table 4. Specifically, we observed that when the number of component values is **fewer than 10**, the search space is too limited, which reduces component diversity. Conversely, when we generate **more than 20** component values, **value redundancy** begins to appear. For example:
>
> - With fewer than 10 values, we obtain candidates like _"Grammar Editor"_.
>
> - With 10 values, we see both _"Grammar Editor"_ and _"Clarity Editor"_.
>
> - With more than 20 values, combinations such as _"Grammar & Clarity Editor"_ start to appear, effectively recombining previously existing values.
>
> Although this expansion increases the size of the search space, the redundancy actually **scatters** the meaningful combinations among valid component values. However, as shown in Table 4, when the number of component values is large, the population's initial performance tends to be stabilized through diverse combinations, keeping the overall variance under control. This indirectly demonstrates that the _component_ concept introduced in our work indeed contributes to performance stability.
>
> **In summary**, based on empirical observations and the analysis above, we recommend practitioners adopt **10** as the initial number of component values, as we do in the paper. Since users come from diverse backgrounds and face domain-specific challenges, they may also increase this number if the optimization results do not meet expectations. Adjusting the hyperparameter **`N`** in Figure 4 allows users to quickly generate a different number of component values.
>
> > (f) **Q3: Clarifying the contribution and utility of individual components**
>
> **Response**: Thank you for this valuable question. We will respond to this question from the following aspects.
>
> **1. Clarification of components.**  The detailed definitions of _components_, _component types_, _component values_, as well as the number of types and values, are provided in **Question 2** for the reviewer’s reference.

---

> ### Author Response · Authors · 2025-11-21
> **Response to Reviewer ga5y (Part 4)**
>
> (cont.)
>
> **2. Ablation and importance of components.**  In general, because tasks vary widely in structure and reasoning requirements, the _optimal_ set of components cannot be universally fixed across all tasks. DelvePO addresses this challenge by introducing the concept of components, which provides users from different domains with a coherent entry point to define task-relevant factors for their specific LLM and task setting. Moreover, to mitigate the risk that certain components may have negative effects during optimization — for instance, the token-accumulation issue previously observed in PromptBreeder — we include a **“null”** option when initializing component values. This allows the framework to effectively suppress or discard irrelevant components during optimization. This mechanism is described in the manuscript at line 933.
>
> To further clarify the reviewer’s concern, we conducted a **simple ablation** by removing each of the five initial components individually and evaluating the resulting impact on performance. The results are shown in the table below:
>
> | abla_type            | Subj   | SQuAD     | SAMSum    |
> | -------------------- | ------ | --------- | --------- |
> | w/o role             | 74.4   | 74.44     | 30.66     |
> | w/o task description | 75.8   | 73.26     | 31.03     |
> | w/o output format    | 72.6   | 75.48     | 30.38     |
> | w/o workflow         | 71.2   | **77.74** | 30.94     |
> | w/o example          | 59.4   | 71.68     | 27.15     |
> | w/ all components    | **76** | 71.78     | **31.74** |
>
> From the table, we can observe that the influence of each component varies across tasks, consistent with the analysis in the manuscript: **different tasks rely on different aspects of the prompt**, and the contribution of each component is therefore task-dependent. The component set we provide in the paper should thus be viewed as a **recommended initial selection**, rather than a universally optimal configuration. And this variability actually highlights an advantage of DelvePO:
>
> - When the user **does not** specify components, the framework provides a reasonable default set.
>
> - When the user **does** have domain knowledge, DelvePO offers a flexible venue for incorporating it.
>
> - The inclusion of the **“null”** option, combined with our optimization framework, enables DelvePO to efficiently converge to task-appropriate prompts without requiring manual tuning.
>
> We hope this clarifies the contribution and utility of individual components within DelvePO.
>
> **3. Performance scaling with number of component types**
> In prompt engineering, the **most critical factor for performance is task-relevant content**, which in DelvePO is represented through component values. Increasing the number of component types can enrich prompt content and potentially improve performance, but it also introduces natural trade-offs:
>
> 1. **Longer input context**, which increases computational cost.
>
> 2. **Slower execution**, as LLMs take more time to process and reason over longer prompts.
>
> To explore this, we increased the number of component types to **10** and conducted experiments on **three datasets** (results shown in the table below).
>
> | # Component Types | SQuAD     | SAMSum    | FinPB    |
> | ----------------- | --------- | --------- | -------- |
> | 5                 | **71.78** | **31.74** | 84.6     |
> | 10                | 69.03     | 29.56     | **85.5** |
>
> We can observe that **two datasets performed better with 5 components**, while **one dataset benefited from 10 components**. This variability likely reflects what we discussed **above**: the **optimal combination of component types depends on task characteristics**, meaning that different tasks may require different aspects of prompt content. Within the DelvePO framework, when users do not specify components, the **preset initial values** are sufficient to achieve effective optimization, as confirmed by our main and supplementary experiments. For users who wish to define their own components, the framework provides a **convenient parameter** to do so (which are detailed in the response to **Question 2** above). Moreover, to address potential issues arising from longer prompts with many components, we include a **“null”** option during component initialization, which effectively neutralizes any components that could negatively impact overall performance (see Appendix E for details).
>
> > (g) **Q4: Clarification and empirical comparison between discrete and continuous prompt memory**
>
> **Response:** Thank you for raising this question. The main distinction between **discrete** and **continuous** prompt memory is as follows:
>
> 1. **Discrete memory** stores the values of components directly, allowing the LLM to independently infer relationships among those values based on its internal reasoning, without relying on an imposed sequence or structure.

---

> ### Author Response · Authors · 2025-11-21
> **Response to Reviewer ga5y (Part 5)**
>
> (cont.)
>
>
> 2. **Continuous memory** arranges the values of components with connecting words, maintaining a specific order (temporal or logical), thereby providing the LLM with an **artificially guided perspective** on how these values relate to one another.
>
>
> **Motivation for the two memory forms.** Prior studies [1, 2] suggest that what LLMs process effectively is not always highly readable text, but often basic token combinations. Based on this, we designed two memory forms to provide LLMs with **flexible ways to leverage the insights derived from the values in prompts**, balancing structure and freedom in reasoning.
>
> To further address your concern, we randomly sampled **four datasets** from our main experiments and conducted **empirical comparisons between discrete and continuous memory**. The results (**Mean values** are reported for DeepSeek-R1-Distill-Llama-8B) are shown in the table below.
>
> | Form of Memory  | Subj      | SQuAD     | FinPB     | SAMSum    |
> | --------------- | --------- | --------- | --------- | --------- |
> | Discrete-Only   | 74.60     | 70.97     | **86.00** | 32.17     |
> | Continuous-Only | 73.90     | **72.69** | 84.30     | **32.83** |
> | All             | **75.90** | 71.78     | 84.60     | 31.74     |
>
> The results indicate that different memory forms contribute differently depending on the dataset, which aligns with our prior explanation. To balance these effects, DelvePO incorporates both memory forms during optimization, effectively integrating their complementary advantages. We hope this explanation clarifies the distinction and the rationale behind our design.
>
> > [1] Lester, Brian, Rami Al-Rfou, and Noah Constant. "The power of scale for parameter-efficient prompt tuning." _arXiv preprint arXiv:2104.08691_ (2021).
> >
> > [2] Liu, Xiao, et al. "GPT understands, too." _AI Open_ 5 (2024): 208-215.
>
> > (h) **Q5: Justification for using different dataset subsets for different LLMs**
>
> **Response:** We thank the reviewer for raising this concern. As the responses to **W2 & Q1** stated before, the experiments on PO is highly resource-intensive. In our dataset selection, we aimed to **balance dataset diversity with computational constraints** (see details under _Experimental Resources and Number of Runs_). Consequently, different subsets of datasets were used for different LLMs, which may have led to the impression of inconsistent model-specific comparisons. We sincerely apologize for any confusion this may have caused.
>
> To address this issue, we conducted **supplementary experiments** on three datasets —**SST-5, FinFE, and AG's News**—  that were not intentionally ignored for DeepSeek-R1-Distill-Llama-8B. The mean values are reported in the table below. From the table, it can be observed that DelvePO still consistently outperforms the baselines on these previously **omitted** datasets. We hope these additional experiments help clarify the misunderstanding to model-specific behaviors and address your concern.
>
> | Baselines     | SST-5     | FinFE     | AG's News |
> | ------------- | --------- | --------- | --------- |
> | APE           | 50.00     | 60.73     | 25.00     |
> | EvoPrompt     | 44.00     | 60.00     | 60.00     |
> | Promptbreeder | 45.00     | -         | 80.00     |
> | DelvePO       | **52.00** | **63.57** | **85.00** |
>
> **Experimental resources and number of runs:** All experiments were conducted on a computing node equipped with **four NVIDIA Tesla V100-SXM2 GPUs (32GB memory each)**. Specifically:
>
> - For each dataset, **non-automated baselines** (Human + CoT with 2 versions) were run for **1 run × 10 epochs × 3 methods**.
>
> - For **automated baselines**, PromptBreeder is relatively time-consuming, so we ran **1 run × 10 epochs × 1 method**. For the other baselines, we ran **3 runs × 10 epochs × 3 methods = 90 epochs**.
>
> In total, for a single dataset, **approximately 130 epochs** were executed (excluding additional evaluations and ablation studies). This setup was designed to **ensure robust results while efficiently managing computational resources**.
>
> > (i) **Q6: How the quality of component initial values affect DelvePO**
>
> **Response:** We thank the reviewer for this question. **Theoretically**, as discussed in our response to **Question 2**, the values of components are ultimately integrated into an individual prompt. Therefore, lower initial quality of component values can have a noticeable impact on individual performance. In DelvePO, however, the component-based design combined with the memory mechanism effectively balances exploration and exploitation, allowing the framework to explore potentially useful variations even when initial component values are of low quality.
>
> To address your question **empirically**, we generated lower-quality initial component values using a smaller LLM and re-executed experiments on two randomly selected datasets for DeepSeek-R1-Distill-Llama-8B. The mean values from one run are shown below:

---

> ### Author Response · Authors · 2025-11-21
> **Response to Reviewer ga5y (Part 6)**
>
> (cont.)
>
>
> | Baselines        | TREC      | SAMSum    |
> | ---------------- | --------- | --------- |
> | APE              | 38.93     | 28.80     |
> | EvoPrompt        | 70.27     | 29.53     |
> | **DelvePO(low)** | 72.90     | 31.50     |
> | DelvePO(org)     | **73.57** | **31.92** |
>
> From these results, we can observe two key points:
>
> 1. Comparing **DelvePO(low)** and **DelvePO(org)**, it is evident that lowering the initial quality of component values does affect performance, which aligns with our analysis above. This is also consistent with classical Evolutionay Algorithm findings, where the quality of the initial population is an important factor.
>
> 2. Comparing **DelvePO(low)** with the baselines, DelvePO still outperforms them despite the reduced initial quality. This demonstrates the effectiveness of DelvePO in balancing exploration and exploitation. Specifically, when component values in selected individuals are suboptimal, DelvePO leverages Prompt Memory to quickly optimize these component values. Conversely, when component values are already strong, DelvePO uses insights from the current Prompt Memory to guide evolutionary variations across the population (Details are provided in our response to **Question 1** mentioned by reviewer afkT).
>
>
> We hope this explanation and the additional results help clarify the robustness of DelvePO with respect to the initial component quality.
>
> > (j) **Q7: Analysis of DelvePO’s failure modes and scenarios of underperformance**
>
> **Response:** We thank the reviewer for this insightful question. During our experiments, we identified a **specific scenario** in which DelvePO exhibits slightly lower performance under GPT-4o-mini. We provide a detailed analysis below.
>
> While analyzing results on dataset **CoLA**, we observed that DelvePO’s performance under GPT-4o-mini was marginally below that of a simpler baseline (**CoT-FS**). This performance difference appears to be related to the dataset version: CoLA has two published variants—one processed with tokenization (**Tokenized**) and one retaining the original sentences (**Raw**). GPT-based models inherently incorporate strong tokenization capabilities, and when prompted with examples similar to CoT-FS, they can produce intermediate reasoning steps effectively using prompts such as _“Step by Step”_. This allows the model to achieve strong performance through straightforward imitation.
>
> In contrast, DelvePO’s prompts involve multiple component values, encouraging GPT-4o-mini to consider the target problem from multiple perspectives. While this design generally improves generality, it can slightly reduce performance on this particular dataset variant. Importantly, your observation highlights a **potential avenue for future work**. In the current experiments, we used a **single set of initial component types and values** without dataset-specific adaptation. Future work could differentiate datasets by complexity and adjust the number and type of components accordingly—for example, by training a **complexity classifier** for each dataset [1].
>
> Nonetheless, it is worth emphasizing that, despite this slight underperformance on CoLA, DelvePO still outperforms other automated PO methods overall, demonstrating its robust advantage within automated prompt optimization frameworks.
>
> > [1] Jeong, Soyeong, et al. "Adaptive-rag: Learning to adapt retrieval-augmented large language models through question complexity." _arXiv preprint arXiv:2403.14403_ (2024).
>
> > (k) **Q8: Clarifying the generality and transferability of DelvePO**
>
> **Response**: Thank you for the question. We understand your concern regarding the **generality and transferability** of our framework across multiple tasks.
>
> The design of the meta-prompts in this work involves two main steps:
>
> 1. **Initial design based on the framework pipeline:** We created the first version of each meta-prompt following the algorithmic flow presented in our framework (see Algorithm 1 for details).
>
> 2. **Calibration using unit-test principles:** After writing the full prompts into the file (`pattern.py`), we verified and calibrated the input-output behavior to ensure correctness and consistency.
>
> **It is important to emphasize that** the meta-prompts, once designed, **do not require further modification**. That is, the **same set of meta-prompts is applied across all datasets**, without any user intervention, demonstrating strong transferability. Differences between tasks are limited to the **task-specific instruction text**, not the meta-prompts themselves.
>
> As reported in the experimental results, our meta-prompts **work well across all datasets**, validating their **robust generalization capability**.
>
> ---
>
> Thanks a lot for your insightful comments and detailed suggestions! We hope our responses address your concerns and enhance your view of our work!

---

### Author Response · Authors · 2025-12-03
**Global Response and Summary of Authors' Responses (Part 1/3)**

Dear **ACs** and **PCs**,

We sincerely appreciate your hard work throughout the review process and for ensuring that our rebuttal was carefully assessed. We are also deeply grateful to all reviewers for their constructive feedback and insightful perspectives. **We are encouraged to see that reviewers recognized**: **(a)** the value of introducing *components* through an analogy to genetic concepts—*Loci* and *Alleles*—which allows prompts to be decomposed into interpretable units, thereby improving the transparency and interpretability of the optimization process (ga5y, akfT); **(b)** the sensibility and effectiveness of the component-based memory mechanism, which makes LLM-driven random mutation more controllable and reusable—reducing LLM stochasticity and making prompt updates more principled and consistent (ga5y, URXt, akfT); **(c)** that our motivation is clear and meaningful, and that the manuscript is clearly written—an especially encouraging recognition from two reviewers who have high confidence on their score (y3qT, akfT); **(d)** the effectiveness and transferability of DelvePO demonstrated across **7 datasets** from **four challenging and practical domains** on **three LLMs** (two open-source, one closed-source) (ga5y, y3qT, akfT).

We carefully read all reviews and have addressed every question raised by the reviewers through additional experiments and thorough clarifications in the individual responses. Among these, we would like to ***summarize the major concerns*** raised by reviewers and ***re-emphasize the core contributions*** of DelvePO in this global response (although most reviewers already found our core contribution clear).

---

### **Summary of Main reviewers' concerns and Our responses**

**(1) Additional experiments on more domains & clarifications on transferability**

**First**, to demonstrate the effectiveness and generalizability of the framework (ga5y, y3qT, URXt), our dataset selection follows two principles:  **1)** wide usage in the PO community, and  **2)** fairness in comparison to baselines. Specifically, we extensively surveyed PO-related work and categorized datasets according to the domains commonly adopted by baselines. Consequently, our main experiments cover **7 datasets across 4 domains**, where DelvePO consistently outperforms all baselines. Given the resource-intensive nature of PO research, we further explored more diverse datasets in two additional experiments, and DelvePO again demonstrated consistent superiority. Following the suggestions from y3qT and URXt, we also incorporated experiments on even more datasets (**GSM8K**, **Casual Judgement**, **Ruin Names**, **Biosses**, **Med QA**), providing stronger evidence of DelvePO’s generalizability.

**Second**, regarding transferability (ga5y, URXt, akfT), reviewers’ concerns mainly centered around: **1)** the design of the meta-prompt and **2)** the use of components. We therefore elaborated on the meta-prompt design process: it is initially constructed according to the algorithmic workflow and then calibrated via unit testing to validate the input and output. Importantly, **the meta-prompt is designed only once** and is used across *all* datasets without any modification from users.

**Besides**, for the introduction of components, we considered two emerging challenges given the rapid development of LLMs:  **1)** users with diverse backgrounds need a PO framework that is both universal and easily customizable with domain knowledge;  **2)** different LLMs exhibit different performance preferences, a trend that will become more pronounced in the future. Inspired by the concept of *Loci* (the corresponding locations of genes with important functions) and *Alleles* (different versions of the same gene) on genetics, we decouple prompt instructions into functional components (analogous to *Loci*) and survey a wide range of component types that may impact performance, ultimately providing two options:  **1)** a *classic* combination used across all our main experiments, and **2)** a *lightweight entry* for user-defined components. All datasets in the main experiments—and the additional datasets suggested by reviewers—were based on the same classic combination, and the **effectiveness** as well as the **transferability** of this design was further validated in our new experiments.

**In summary**, beyond the experiments initially reported in the manuscript, the newly added experiments consistently show superior performance compared to all baselines. This not only confirms DelvePO’s effectiveness but also provides strong evidence of its transferability. As LLM usage continues to expand across diverse fields, the novel component mechanism of DelvePO will make the framework increasingly flexible and widely adopted.

---

**(2) Cost-related concerns: clarification of cost differences & new experiments on performance vs. budget**

We clarified **3** key points regarding cost:

---

> ### Author Response · Authors · 2025-12-03
> **Global Response and Summary of Authors' Responses (Part 2/3)**
>
> (cont.)
>
>
> **First**, a typo in the caption of Table 6 unintentionally suggested a larger cost gap than what actually exists. Once corrected, the true difference between DelvePO and baselines turns out to be minimal and well within the expected range.
>
> **Second**, across all three LLMs (two open-source and one closed-source), DelvePO consistently surpasses baselines on both performance and time efficiency for open-source models. For the closed-source model, DelvePO delivers higher—or at least comparable—performance while keeping cost at a similar and competitive level, demonstrating that stronger results do not arise from excessive token usage.
>
> **Third**, following the suggestions from reviewers **y3qT**, **URXt**, and **ga5y**, we conducted additional experiments under both **fixed budgets** and **varying budget levels**. These experiments yielded two key insights:
>
> - relying solely on iteration count (as done in classic PO) is insufficient; optimization should instead consider **both** iteration thresholds **and** budget constraints;
>
> - evaluating a PO framework requires jointly assessing performance and cost (time and tokens), enabling a more complete understanding of efficiency.
>
>
> These analyses are discussed in detail in our response to **Question 2** from reviewer **y3qT**. We truly appreciate the reviewers’ suggestion to examine performance under fixed budgets—it substantially strengthened our evidence for DelvePO’s efficiency.
>
> **Overall**, our findings show that:
> **1)** On **open-source LLMs**, DelvePO consistently outperforms baselines in both speed (time) and final performance.
> **2)** On the **closed-source LLM**, DelvePO converges faster **under the same budget**, reaches better optima earlier than baselines; and continues to improve steadily **under various budgets**. These results indicate that DelvePO maintains superior optimization efficiency across practical cost settings. Based on the above clarifications, we can anticipate that these additional experiments and detailed explanations could sufficiently address reviewers' concerns and further enhance their view of our work!
>
> ---
>
> **(3) Additional explanations of technical details**
>
> Beyond the two major concerns shared across reviewers, several questions focused on technical implementation details. Many of these points—such as the choice of component number, the effect of initialization values, and clarifications of certain concepts—were already addressed in the manuscript, but due to the limited pages, most fine-grained explanations had to be relegated to the appendices. Nevertheless, to more clearly respond to each reviewer’s specific concerns, we revisited all related comments, conducted additional empirical experiments where appropriate, and provided expanded explanations in the corresponding responses. Interestingly, most detailed questions were raised by reviewer ga5y. While ga5y may not be deeply familiar with the PO literature, their comments clearly show strong curiosity and engagement with our work. To ensure clarity and completeness, we prepared **six dedicated response windows** for these comments, offering detailed discussions and extensive supplemental experiments. We believe these thorough responses can effectively resolve the confusions raised by ga5y.
>
> We also sincerely appreciate reviewer **URXt** for offering valuable insights into further improving DelvePO—specifically, by identifying stages of the framework that could potentially be parallelized or cached to reduce computational overhead. While such system-level exploration is beyond the scope of this paper, it represents a promising avenue for future research, especially as DelvePO becomes increasingly adopted for diverse real-world tasks. We are committed to exploring these promising directions and sharing our progress with the community.
>
> ---
>
> **To sum up**, except for the first two major concerns above, most other questions arose because the space limitations compressed our explanations of technical details, which inevitably made it more time-consuming for reviewers who are unfamiliar with PO to understand our design, potentially leading to confusion or misinterpretation. After the detailed clarifications throughout the rebuttal, we believe these concerns have been effectively resolved.
>
> We fully understand the busy schedules of **ACs**, **PCs**, and **reviewers**. To facilitate a quick understanding of our work and the overall rebuttal process, we summarize the main efforts made during the rebuttal and kindly direct ACs and PCs to the corresponding responses. We would also like to gently remind reviewers to check the **new figures/files** provided via **[anonymous links](https://anonymous.4open.science/r/ICLR-DelvePO-Rebuttal)** embedded in the text (as they cannot be directly inserted into the response windows).

---

> > ### Author Response · Authors · 2025-12-03
> > **Global Response and Summary of Authors' Responses (Part 3/3)**
> >
> > (cont.)
> >
> >
> > - Following ga5y, URXt, akfT: clarified transferability and generalizability of DelvePO;
> > - Following ga5y, y3qT, URXt: conducted new experiments under the same budget and various budgets;
> > - Following ga5y, y3qT, URXt: clarified the principle of dataset selection and conducted additional experiments on more challenging and difficult tasks in LLM-era;
> > - Following ga5y, y3qT, akfT: clarified the organization of Case Study and provided more detailed examples;
> > - Following ga5y: clarified the contribution, utility and configuration of components (including types and values);
> > - Following ga5y: conducted new experiments on different form of Prompt Memory;
> > - Following akfT: added new methods into related work and conducted new experiments on closer, up-to-date baseline;
> > - Following URXt: added discussions about which stages can be parallelized and which stages can be cached in advance;
> > - Following y3qT and URXt: clarified several concepts and corrected some typos.
> >
> > ---
> >
> > ### **Re-emphasize the Contributions of DelvePO**
> >
> > **First**, this paper addresses several limitations commonly found in current Prompt Optimization (PO) methods, including **rigid and inflexible optimization frameworks**, **slow optimization processes**, and **poor interpretability**. To overcome these issues, we introduce **DelvePO**, a novel framework built upon evolutionary algorithms. Inspired by the concepts of **Loci** and **Alleles** in genetics, we propose the notion of **components**, which substantially enhances the flexibility and interpretability of the optimization workflow. This design allows users with diverse domain backgrounds to solve specialized tasks more conveniently and efficiently. In addition, building on components, we incorporate **diversified working-memory mechanisms**, which enable the reuse of intermediate optimization states and make the evolutionary process more controllable.
> >
> > **Second**, we conduct extensive experiments across multiple domains and on both open-source and closed-source LLMs, and the results show that DelvePO **consistently outperforms all baselines**. Specifically, on open-source models, DelvePO achieves superior performance and lower time consumption. On closed-source models, DelvePO still surpasses baselines in terms of final performance. Regarding computational cost—which was a major concern raised by the reviewers—we followed their valuable suggestions and performed additional experiments under various budget settings. The findings demonstrate that: (1) **under the same budget**, DelvePO converges more rapidly to optimal performance; and (2) **under different budgets**, DelvePO not only achieves the best performance but also maintains a more stable optimization process.
> >
> > **Last but not least**, to the best of our knowledge, we are the **first** to **formally introduce and study the concepts of Components and Working Memory** in the context of Prompt Optimization. As highlighted in our response to ga5y-W1 concerning the core contributions of our work, although employing an evolutionary algorithm, DelvePO is **not** a simple incremental improvement. Instead, it introduces a **new perspective** that opens the door to a fast, interpretable, and flexible framework for automated PO—where evolutionary algorithms are just one of the potential instantiations we have explored in our current research. We believe this direction is promising for future PO research and will significantly lower the barrier for non-AI experts to leverage LLMs effectively.
> >
> > As more powerful LLMs emerge with the ability to handle longer contexts, we anticipate that diverse, domain-specific prompts will become increasingly essential across many fields. We firmly believe that DelvePO will help a wider range of users complete complex tasks with greater efficiency and reliability.
> >
> > Thank you once again for your careful consideration and for the attention you have devoted to our work. We also sincerely appreciate the reviewers for their thoughtful and insightful comments. All additional experiments, analyses, and suggestions will be carefully incorporated into the revised manuscript.
> >
> > **Best Regards**,
> > All authors of submission **15620** (**DelvePO**)

---

### Meta-Review · Area_Chair_Y53k · 2026-01-07

**Summary:**

This paper proposes DelvePO, a framework for automatic prompt optimization that decomposes prompts into modular components and uses a working memory mechanism to guide evolutionary operations. The method is evaluated across multiple LLMs and datasets, demonstrating improvements over several evolutionary prompt optimization baselines.

The paper presents a technically sound extension of evolutionary prompt optimization. However, the core contributions are perceived as incremental, and several significant methodological and presentational concerns raised by reviewers remain insufficiently addressed, placing the paper marginally below the acceptance threshold.

**Reviewer Concerns:**

I thank the authors have made great efforts to address each reviewer's comments in detail.

However, as multiple reviewers indicated, the core idea of componentization and memory-augmented evolution to be a clear but incremental advance over existing evolutionary optimization paradigms.

A further question pertains to the practical deployment of this method. The proposed optimization procedure appears quite complex, raising concerns about its real-world adoption. It is worth considering whether the performance gains justify the significant overhead, as few end-users of LLMs would likely engage with such an intricate process for marginal improvements."

**Reviewer Scores:**

I think the orignal ratings of the first three reviewers might slight change, but the overall rating I think is still boderline reject.

---

### Decision · Program_Chairs · 2026-01-26

Reject